# Increased anxiety and decreased sociability induced by paternal deprivation involve the PVN-PrL OTergic pathway

Zhixiong He[1], Larry Young[2,3], Xin-Ming Ma[1,4], Qianqian Guo[1], Limin Wang[1], Yang Yang[1], Luo Luo[1], Wei Yuan[1], Laifu Li[1], Jing Zhang[1], Wenjuan Hou[1], Hui Qiao[1], Rui Jia[1], Fadao Tai[1]*

[1]Institute of Brain and Behavioral Sciences, College of Life Sciences, Shaanxi Normal University, Xi'an, China; [2]Department of Psychiatry and Behavioral Sciences, Silvio O. Conte Center for Oxytocin and Social Cognition, Center for Translational Social Neuroscience, Yerkes National Primate Research Center, Emory University, Atlanta, United States; [3]Center for Social Neural Networks, University of Tsukuba, Tsukuba, Japan; [4]Department of Neuroscience, University of Connecticut Health Center, Farmington, United States

**Abstract** Early adverse experiences often have devastating consequences. However, whether preweaning paternal deprivation (PD) affects emotional and social behaviors and their underlying neural mechanisms remain unexplored. Using monogamous mandarin voles, we found that PD increased anxiety-like behavior and attenuated social preference in adulthood. PD also decreased the number of oxytocin (OT)-positive neurons projecting from the paraventricular nucleus (PVN) and reduced the levels of the medial prefrontal cortex OT receptor protein in females and of the OT receptor and V1a receptor proteins in males. Intra-prelimbic cortical OT injections reversed the PD-induced changes in anxiety-like behavior and social preferences. Optogenetic activation of the prelimbic cortex OT terminals from PVN OT neurons reversed the PD-induced changes in emotion and social preference behaviors, whereas optogenetic inhibition was anxiogenic and impaired social preference in naive voles. These findings demonstrate that PD increases anxiety-like behavior and attenuates social preferences through the involvement of PVN OT neuron projections to the prelimbic cortex.
DOI: https://doi.org/10.7554/eLife.44026.001

*For correspondence:
taifadao@snnu.edu.cn

Competing interests: The authors declare that no competing interests exist.

## Introduction

Social attachments can facilitate reproduction, increase survival, provide a sense of security and reduce feelings of stress and anxiety in many species (*Coria-Avila et al., 2014*). In humans, attachment is especially important during early development because disruption of filial attachment in children (e.g., abuse, neglect or the death of a parent) increases their vulnerability to emotional disorders at a later age (*Agid et al., 1999*; *Reinherz et al., 1999*).

By the end of 2013, it was estimated that over 61 million children in rural hometowns were missing one or both parents (more often the father) throughout China (*All-China Women's Federation, 2013*). Neonatal social or paternal deprivation (PD) has been proven to exert a profound and persistent influence on the physiological and behavioral development of offspring. Neonatal rodents physiologically and emotionally depend on their parents. For example, PD impairs sociability (*Bambico et al., 2015*; *Jia et al., 2009*) and social recognition (*Cao et al., 2014*), reduces parental

**eLife digest** Parental care early in life is essential for normal development of the brain in humans and some other animals. It also lays the ground work for healthy behaviors later in life. Many studies have looked at the importance of a mother's care, but less attention has been paid to the role played by fathers. Research shows that children who grow up without a father are at risk of emotional and behavioral problems later in life. But it is not clear how missing a father's care affects brain development.

Oxytocin, a chemical produced by a part of the brain called the paraventricular nucleus, plays a key role in parental bonding. Another part of the brain called the prelimbic cortex regulates many emotions and many complex behaviors. Studying animals, like the mandarin vole, that form strong bonds with both parents is one way to learn more about how the loss of paternal care affects oxytocin or emotional and behavioral health.

Now, He et al. show that mandarin voles raised without a father are more anxious and socialize less with other voles than those raised with a father. The voles deprived of paternal care also have fewer oxytocin-producing cells in the paraventricular nucleus and fewer receptors for oxytocin in the prelimbic cortex. Injecting oxytocin into the prelimbic cortex eliminated the anxious and antisocial behavior seen in the voles lacking paternal care. Using a technique called optogenetics to restore the release of oxytocin in the prelimbic cortex reduced anxious behavior and restored normal social interactions. Using the same approach to interfere with communication between the paraventricular nucleus and prelimbic cortex in voles raised with a father also triggered anxious and antisocial behavior.

The experiments reveal that fathers play an important role in brain and behavioral development in mandarin voles. He et al. show that a lack of paternal care leads to deficits in oxytocin and a poor communication between the paraventricular nucleus and prelimbic cortex that contribute to emotional and social abnormalities in the voles. More studies are needed to determine father's care has similar effects in humans. But if this relationship is confirmed, it might lead scientists to develop new strategies for treating psychiatric disorders in people deprived of paternal care.
DOI: https://doi.org/10.7554/eLife.44026.002

behaviors (*Jia et al., 2009*) and alloparental behavior (*Ahern and Young, 2009*), and inhibits the formation of pair bonding (*Yu et al., 2013*). Lack of paternal care can also affect emotional behavior (*Ovtscharoff et al., 2006*). In humans, PD impairs psychological and mental development and increases the risk of substance abuse and personality disorders (*Grossmann et al., 2002*; *Jablonska and Lindberg, 2007*; *Sobrinho et al., 2012*). Despite emerging evidence on the impact of PD, the primary focus of research has been on father–offspring relationships from postnatal day (PND) 0–21 in rodents (*Gos et al., 2014*; *Ahern and Young, 2009*).

Mandarin voles (*Microtus mandarinus*) are socially monogamous and exhibit extensive biparental investment and high offspring survival and growth (*Tai et al., 2001*; *Tai and Wang, 2001*). Mandarin voles are furred by around PND 7, suggesting that their thermoregulatory ability is well developed. Mandarin voles open their eyes and begin to eat solid food around PND 13. Paternal care (e.g., licking, retrieving and nest building) gradually decreases from PND 14 to PND 20 (*Wang et al., 2018*). Pups of age PND 1–13 and PND 14–21 have different needs, and the effects of disrupting father–offspring bonds during the latter period may not be the result of disrupted direct care but the result of disrupted emotional attachment (*He et al., 2017*). Mandarin vole pups have high levels of attachment to their fathers from PND 14–21 (*He et al., 2017*). Whether PD from PND 14–21 affects emotional and social behaviors and their neuroendocrine mechanisms remain unexplored.

The neuropeptide oxytocin (OT) is primarily produced in neurons of the hypothalamic paraventricular nucleus (PVN) and supraoptic nucleus (*Onaka, 2004*). OT is strongly implicated in prosocial behavior (*Marlin et al., 2015*; *Young and Wang, 2004*; *Burkett et al., 2016*; *Oettl et al., 2016*; *Wircer et al., 2017*) and in decreased anxiety-related behavior (*Smith and Wang, 2014*; *Sabihi et al., 2014b*). Previous studies have found that PD alters the levels of OT receptor (OTR) mRNA in the brain (*Cao et al., 2014*), and neonatal OT treatments have long-term effects on behavior and physiology in mandarin voles (*Jia et al., 2009*). OT binds to OTRs (*Burkett et al., 2016*) or

vasopressin 1a receptors (V1aR) (*Song et al., 2014*) to affect social behavior. If preweaning PD changes sociability and emotion, we predict that it should also affect the levels of OT and OTR.

OTRs and V1aRs are found in the medial prefrontal cortex (mPFC) (*Smeltzer et al., 2006*; *Lieberwirth and Wang, 2016*). Several studies suggest that the mPFC is critical for the expression of anxiety-like behavior (*Lisboa et al., 2010*; *Saitoh et al., 2014*; *Wang et al., 2015*) and social behavior (*Sabihi et al., 2014a*; *Lee et al., 2016*; *Murugan et al., 2017*). This brain region is a het-erogeneous cortical structure composed of subregions, including the anterior cingulate (Cg), prelim-bic cortex (PrL) and infralimbic cortex (*Heidbreder and Groenewegen, 2003*). Several studies have shown the involvement of the PrL in the regulation of anxiety (*Sabihi et al., 2014b*; *Wang et al., 2015*) and social behavior (*Young et al., 2001*; *Carrier and Kabbaj, 2012*). A recent report found that a subset of mPFC neurons elevated discharge rates when approaching a strange mouse but not when approaching non-social objects (*Lee et al., 2016*), indicating the involvement of the mPFC in social behavior. However, whether OT in the PrL is involved in the effects of preweaning deprivation on emotion and sociability remains unclear. In addition, whether the PVN-mPFC OTergic neural cir-cuit mediates mood and social behavior has not been determined. Therefore, we investigated whether optogenetic activation of the PVN (OTergic)-mPFC projections can rescue the changes in emotion and social approach-avoidance induced by PD to reveal the involvement of PVN OTergic projections in these processes.

Using socially monogamous mandarin voles, we investigated the effects of PD from PND 14–21 on emotion and social preference and the levels of OT and OTR in specific brain regions. We then tested whether a microinjection of OT into the PrL and optogenetic activation of PVN (OTergic)-mPFC projections can recover the effects of preweaning PD. We hypothesized that disruption of early emotional attachment between pups and fathers affects anxiety-like behavior and social prefer-ence in mandarin voles in adulthood and that the OT system is probably involved in this process.

## Results

### Effect of pre-weaning PD on anxiety-like behavior and social preference

It has previously been shown that offspring that experience neonatal maternal separation or early deprivation display high levels of anxiety-like behavior (*Sachs et al., 2013*; *Wei et al., 2010*; *Koe et al., 2016*). Neonatal adversity has been shown to induce changes in social behavior, includ-ing avoidance, fear and decreased social interactions (*Giachino et al., 2007*; *Jia et al., 2009*; *Toth and Neumann, 2013*). This study was designed to test the hypothesis that preweaning PD (PND 14–21) increases anxiety-like behavior and reduces social preference.

An open field test (OFT) showed that the percentage of time spent in the central area was greater in the biparental care (PC) group than in the PD group (male: t(12) = 4.158, p<0.01, *Figure 1A*, *Figure 1—source data 1*; female: t(12) = 3.226, p<0.05, *Figure 1B*, *Figure 1—source data 1*). However, the total distance covered by voles the from PC group was not different than that covered by voles in the PD group (*Figure 1C–F*, *Figure 1—source data 1*). In the light and dark test, PC voles spent more time in the light compartment than did PD voles (male: t(12) = 2.635, p<0.05, *Figure 1G and I*, *Figure 1—source data 1*; female: t(12) = 3.937, p<0.01, *Figure 1H and J*, *Figure 1—source data 1*).

Two-way ANOVA indicated a significant interaction between the treatment and absence/pres-ence of a stimulus mandarin vole on the percentage of investigation time in males (F(1,24) = 4.775, p<0.05). The males of the PC group engaged in more investigations of the social stimulus than of the object stimulus (p<0.025) (*Figure 1K*, *Figure 1—source data 1*). Preweaning PD caused a decrease in social investigations by males during the social preference test (p<0.025) (*Figure 1K*, *Figure 1—source data 1*). No significant treatment x stimulus interaction was found on the percent-age of investigation time in females (F(1,24) = 1.553, p=0.225), but a main effect of the absence/presence of a social stimulus on mandarin voles was found (F(1,24) = 11.785, p<0.01), such that PC group females engaged in more investigations of the social stimulus than of the object stimulus (p<0.025) (*Figure 1L*, *Figure 1—source data 1*). Treatment did not produce a main effect (F(1,24) = 3.064, p=0.093) in females. There was also no significant interaction in females (F(1,24) = 1.553, p=0.225). PD did not affect investigations of the social stimulus and object stimulus in males (*Figure 1K*, *Figure 1—source data 1*) or females (*Figure 1L*, *Figure 1—source data 1*).

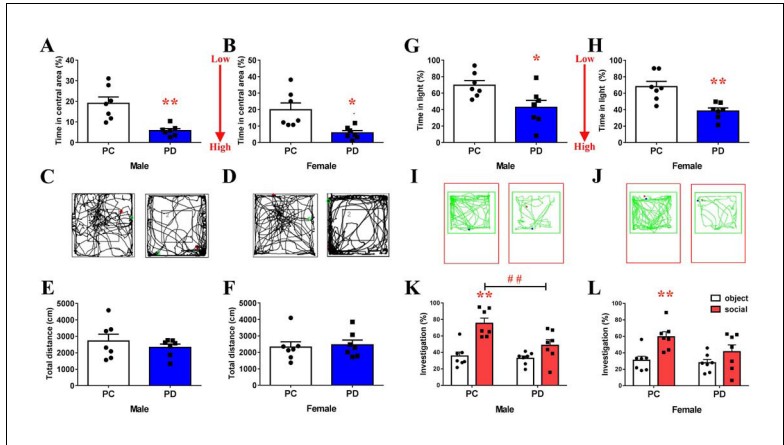

**Figure 1.** Effect of paternal deprivation on anxiety-like behavior and social preference in adult mandarin voles (n = 7). (**A, B**) Percentage of time in the central area, (**C, D**) representative path and (**E, F**) total distance of mandarin voles in the open field test. *p<0.05; **p<0.01. Independent sample t-tests. (**G, H**) Percentage of time in the light area and (**I, J**) animal traces of mandarin voles in the light and dark box. *p<0.05; **p<0.01. Independent sample t-tests. Effect of PD on social preference in (**K**) males and (**L**) females. Error bars indicate SEM. **p<0.025 vs. object stimulus. ##p<0.025 vs. PC. Two-way ANOVA (factors: treatment × stimulus). PC, biparental care; PD, paternal deprivation.

DOI: https://doi.org/10.7554/eLife.44026.003

The following source data is available for figure 1:

**Source data 1.** Statistical results of the percentage of time spent in the central area and total distance in the open field, percentage of time in the light area in the light-dark box, and the percentage of time in investigating the social stimulus or object stimulus in the social preference test.

DOI: https://doi.org/10.7554/eLife.44026.004

## Effect of preweaning PD on the PrL, NAc and PVN neuronal activities in the social-approach test

Recently, a study found that the activity of some mPFC neurons in a test mouse increased during the approach to a stranger mouse but not during the approach to an inanimate object (*Lee et al., 2016*). The amount of c-fos expression increased after encountering a stimulation in the NAc shell but not in the NAc core (*Ago et al., 2013*). PVN OT neuronal activity was enhanced during social interactions (*Hung et al., 2017*). Oxytocinergic neurons within the PVN project to the mPFC (*Knobloch et al., 2012*) and the NAc (*Ross et al., 2009*). We hypothesize that PD alters c-fos expression in the PrL and NAc and reduces the percentage of c-fos/OT double-labeled neurons in the PVN of mandarin voles after exposure to a juvenile vole compared to exposure to a toy.

Subjected were used to exposure to both a juvenile vole (*Figure 2A*) and a toy (*Figure 2B*). The PC group spent more time in investigations of a juvenile vole than a toy stimulus (both males (p<0.025) and females (p<0.025)), whereas there were no differences in the length of investigations when PD voles approached a juvenile vole or a toy (either sex) (*Figure 2—figure supplement 1*). There was a significant interaction effect of treatment (PC versus PD) by stimulus (object versus social target) on the percentage of neurons double-labeled for OT and c-fos (PVN) in both males (F (1,16) = 5.122, p<0.05) and females (F(1,16) = 10.545, p<0.01). The post-hoc test indicated that PC voles had an increased percentage of c-fos/OT-labeled cells after approaching a juvenile vole compared to approaching a toy (in both males (p<0.01) (*Figure 2E*, *Figure 2—source data 1*) and females (p<0.01) (*Figure 2F*, *Figure 2—source data 1*)). By contrast, the percentage of c-fos/OT double-labeled neurons present after approaching a juvenile vole versus after approaching a toy did not significantly differ within PD in males (*Figure 2E*, *Figure 2—source data 1*) or females (*Figure 2F*, *Figure 2—source data 1*). Preweaning PD only reduced the percentage of c-fos/OT-labeled neurons after approaching a juvenile vole in females (p<0.01) (*Figure 2F*, *Figure 2—source data 1*).

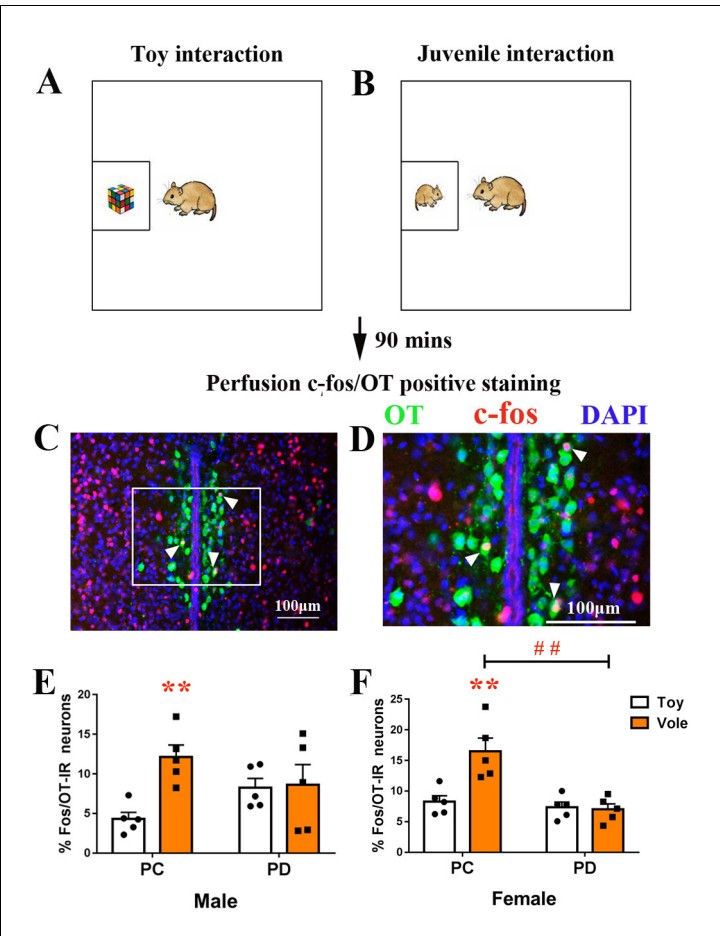

**Figure 2.** Effect of paternal deprivation on the percentage of c-fos/OT double-labeled neurons in the PVN in adult mandarin voles (n = 5). Voles were subjected to interaction with either (**A**) a juvenile voles or (**B**) a magic cube. (**C, D**) Double-immunohistochemical staining of c-fos (red) and OT (blue). Effect of PD on the percentage of neurons double-labeled for OT and c-fos in (**E**) males and (**F**) females. Error bars indicate SEM. **p<0.01 vs. object stimulus. ##p<0.01 vs. PC. Two-way ANOVA (factors: treatment × stimulus). PC, biparental care; PD, paternal deprivation.

DOI: https://doi.org/10.7554/eLife.44026.005

The following source data and figure supplement are available for figure 2:

**Source data 1.** Numbers of c-fos/OT double-labeled positive cells in the PVN.
DOI: https://doi.org/10.7554/eLife.44026.007

**Figure supplement 1.** Paternal deprivation diminishes social approach in (**A**) male and (**B**) female mandarin voles (n = 5).
DOI: https://doi.org/10.7554/eLife.44026.006

After exposure to a juvenile vole or a toy (*Figure 3A*), the brain tissues of subjects were stained for c-fos in the PrL (*Figure 3B*) and the NAc (*Figure 3C*). Two-way ANOVA showed a significant treatment x stimulus interaction on c-fos expression within the PrL in both males ($F_{(1,16)} = 6.831$, p<0.05) and females ($F_{(1,16)} = 7.136$, p<0.05). The PC group had more positive c-fos-labeled neurons after exposure to a juvenile vole than after to exposure to a toy in both males (p<0.01) (*Figure 3D*, *Figure 3—source data 1*) and females (p<0.01) (*Figure 3E*, *Figure 3—source data 1*). However, the PD group showed no effect of simulation type on c-fos expression in neurons in the PrL in either sex. When compared to the PC group, the PD group also had reduced c-fos expression in males (p<0.01) (*Figure 3D*, *Figure 3—source data 1*) and females (p<0.01) (*Figure 3E*, *Figure 3—source data 1*).

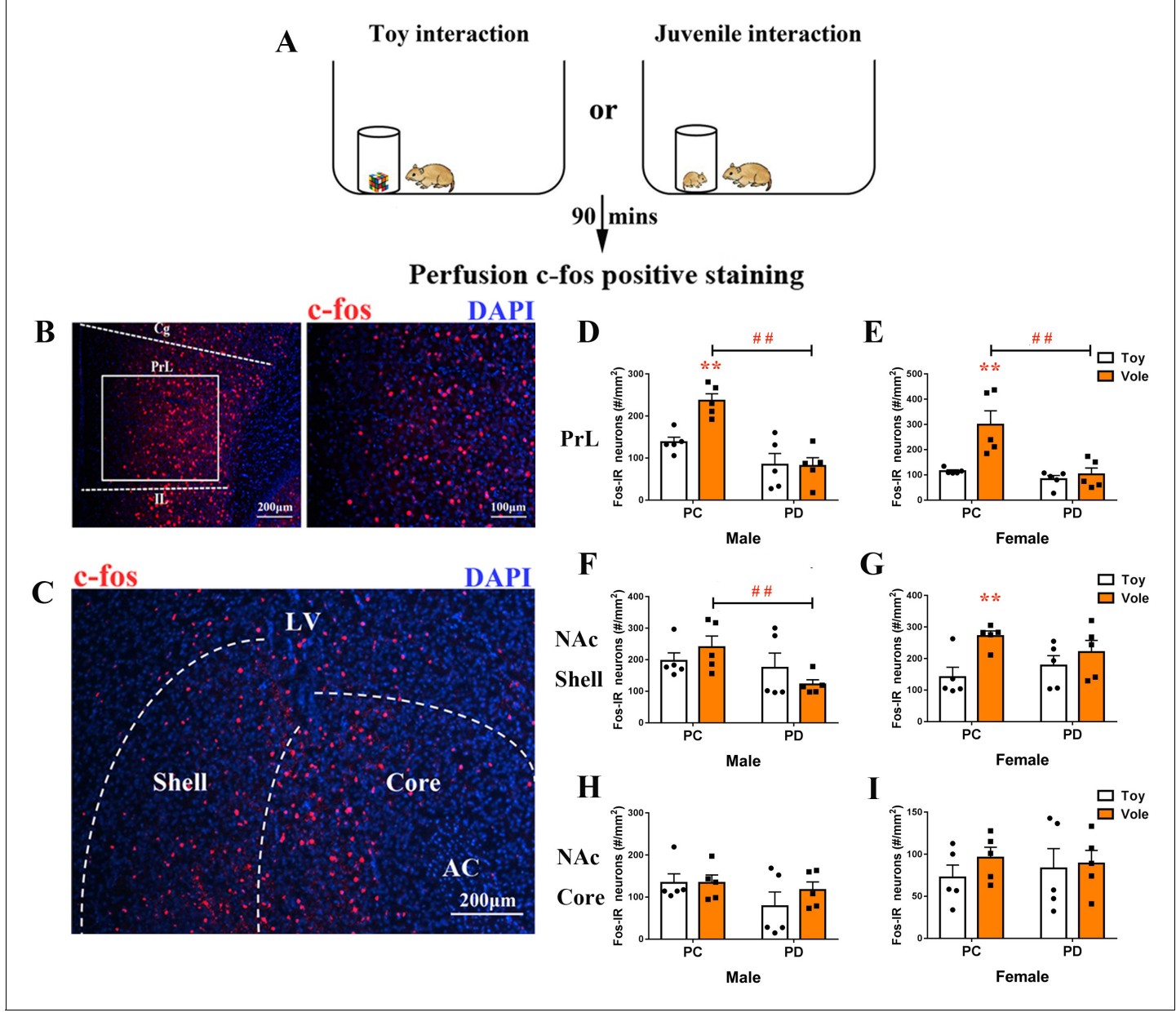

**Figure 3.** Effect of paternal deprivation on c-fos expression in the PrL and the NAc of adult mandarin voles (n = 5). (A) Voles were subjected to interaction with either a juvenile vole or a magic cube. (B) Immunohistochemical staining of c-fos (red) and 4',6-diamidine-2'-phenylindole dihydrochloride (DAPI) (blue). The effect of PD on c-fos expression in the PrL of (D) males and (E) females. (C) Images show c-fos immunoreactivity in the NAc shell and the NAc core. Effect of PD on c-fos expression in (F) males and (G) females of the NAc shell. Effect of PD on c-fos expression in the NAc core of (H) males and (I) females. Error bars indicate SEM. **p<0.01 vs. object stimulus. ##p<0.01 vs. PC. Two-way ANOVA (factors: treatment × stimulus). PC, biparental care; PD, paternal deprivation.

DOI: https://doi.org/10.7554/eLife.44026.008

The following source data is available for figure 3:

**Source data 1.** Numbers of c-fos-positive cells in the PrL, NAc shell and NAc core.

DOI: https://doi.org/10.7554/eLife.44026.009

The sex x treatment interaction had no significant effect on c-fos expression in the NAc shell (males: $F_{(1,16)}$ = 2.209, p=0.157; females: $F_{(1,16)}$ = 2.156, p=0.161) or the NAc core (males: $F_{(1,16)}$ = 0.629, p=0.439; females: $F_{(1,16)}$ = 0.281, p=0.604). Treatments (PC or PD) produced a significant main effect on c-fos expression in the NAc shell in males ($F_{(1,16)}$ = 4,596, p<0.05), but no main eect

was induced by a stimulus (F(1,16) = 0.018, p=0.896). After exposure to a juvenile vole, PD males exhibited decreased c-fos expression in the NAc shell when compared with PC males (p<0.05) (*Figure 3F*, *Figure 3—source data 1*). Stimulus type had a significant main effect on c-fos expression in the NAc shell in females (F(1,16) = 8.422, p<0.01), but PD had no significant effect on c-fos expression when compared to PC (F(1,16) = 0.058, p=0.813). PC females exhibited significantly fewer c-fos-positive nuclei in the NAc shell after approaching a juvenile vole than after approaching a toy (p<0.01), but c-fos expression in neurons was not affected in PD females (*Figure 3G*, *Figure 3—source data 1*). NAc core c-fos expression did not differ between the PC and PD groups for either sex (*Figure 3H and I*, *Figure 3—source data 1*).

## Effect of preweaning PD on PVN OT-IR

Previous studies have shown that neonatal isolation or maternal separation results in a decrease in OT immunoreactive (OT-IR) neurons in the PVN in male adult mandarin voles and female mice (*Wei et al., 2013*; *Veenema et al., 2007*). It is apparent that OT production in the hypothalamus is altered in response to social interactions in many species (*Leng et al., 2008*). Tactile stimulation at an early developmental stage induces immediate-early gene activity in OT neurons in prairie voles (*Barrett et al., 2015*) and rabbit pups (*Caba et al., 2003*), reduces stress responses during adulthood (*Liu et al., 1997*), and alleviates the negative effects of neonatal isolation on novel object recognition and sociability (*Wei et al., 2013*). We therefore tested the number of OT-IR neurons (*Figure 4A-D*) in the PVN (*Figure 4E*).

Two-way ANOVA revealed interactions between treatment and sex in PVN OT-IR neurons (F(1,12) = 8.265, p<0.05). The results of post-hoc tests indicated that males and females in the PD group had fewer PVN OT-IR neurons than males and females in the PC group (both p<0.01) (*Figure 4F*, *Figure 4—source data 1*). Females had more PVN OT-IR neurons (p<0.01) than males (*Figure 4F*, *Figure 4—source data 1*).

## Effect of preweaning PD on mPFC OTR-IR, V1aR-IR and AVP-IR levels

The OT and arginine vasopressin (AVP) peptides and the OTR and V1aR receptors display a high degree of sequence homology, and both peptides can activate both receptors (*Chini and Manning, 2007*). OTR (*Li et al., 2016*), V1aR and AVP (*Dumais and Veenema, 2016*) are involved in the regulation of mood and social behavior. We hypothesized that preweaning PD alters the density of OTR, V1aR and/or AVP in the mPFC, NAc and/or PVN.

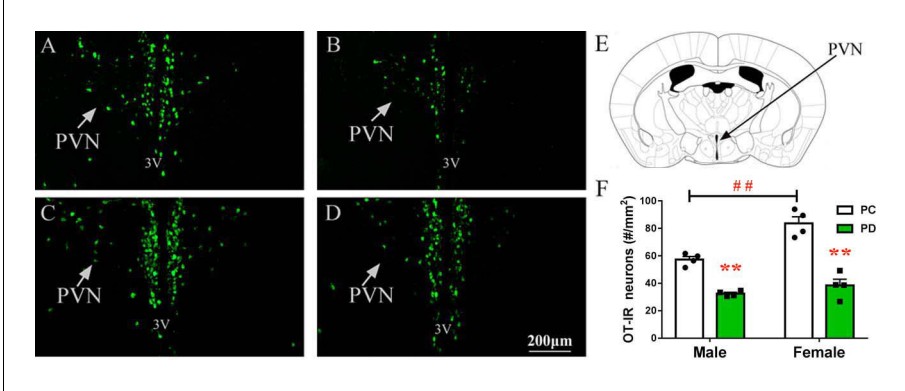

**Figure 4.** Effect of paternal deprivation on PVN OT-IR neurons. (**A**) PC males, (**B**) PD males, (**C**) PC females and (**D**) PD females. 3V, 3rd ventricle. (**E**) Schematic drawing illustrating tissue in the PVN. (**F**) Quantification of OT-IR neurons in the PVN. Error bars indicate SEM. n = 4. **p<0.01 vs. PC. ##p<0.01 vs. male. Two-way ANOVA (factors: treatment × sex). PC, biparental care; PD, paternal deprivation; PVN, paraventricular nucleus.
DOI: https://doi.org/10.7554/eLife.44026.010

The following source data is available for figure 4:

**Source data 1.** Numbers of OT-positive cells in the PVN.
DOI: https://doi.org/10.7554/eLife.44026.011

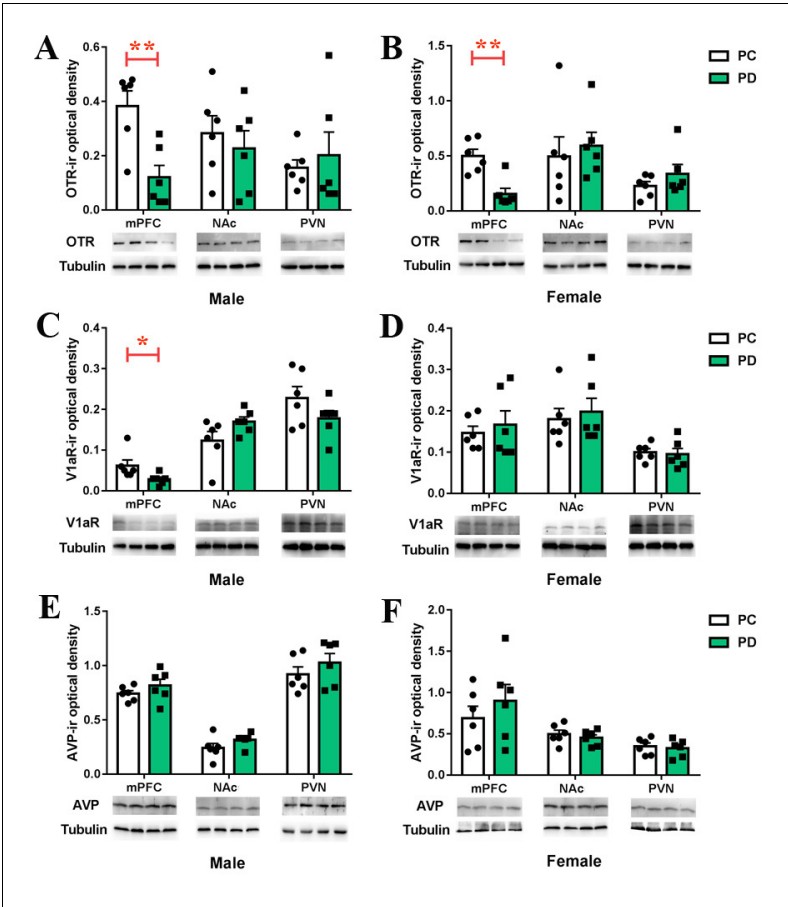

**Figure 5.** Effects of paternal deprivation on mesocorticolimbic. (A–B) OTR, (C–D) V1aR and (E–F) AVP immunoreactivity in male (n = 6) and female (n = 6) mandarin voles. Error bars indicate SEM. *p<0.05; **p<0.01. Independent sample t-tests. AVP, arginine vasopressin; mPFC, medial prefrontal cortex; NAc, nucleus accumbens; OTR,oxytocin receptor; PC, biparental care; PD, paternal deprivation; PVN, paraventricular nucleus; V1aR, vasopressin 1a receptor.

DOI: https://doi.org/10.7554/eLife.44026.012

The following source data is available for figure 5:

**Source data 1.** Levels of OTR, V1aR or AVP in the mPFC, NAc and PVN measured by Western Blot assay.

DOI: https://doi.org/10.7554/eLife.44026.013

PD-group males had lower levels of OTR (t(10) = 3.813, p<0.01) (*Figure 5A*, *Figure 5—source data 1*) and V1aR (t(10) = 2.375, p<0.05) (*Figure 5C*, *Figure 5—source data 1*) proteins in the mPFC than PC-group males. No group differences were observed for the levels of OTR and V1aR proteins in the NAc or PVN in males (*Figure 5A and C*, *Figure 5—source data 1*). PD-group females had lower levels of the OTR protein in the mPFC than PC-group females (t(10) = 4.315, p<0.01) but not in the NAc or PVN (*Figure 5B*, *Figure 5—source data 1*). No group differences were noted for the levels of the V1aR protein in the mPFC, NAc or PVN in females (*Figure 5D*, *Figure 5—source data 1*). PD had no effect on the AVP neuropeptide in the mPFC, NAc or PVN, regardless of sex (*Figure 5E and F*, *Figure 5—source data 1*).

## Effect of microinjection of OT into the PrL on the anxiety-like behavior and social preference altered by preweaning PD

Infusion of OT into the PrL region of the mPFC reduces anxiety-like behavior (*Sabihi et al., 2014b*). OTRs are important for modulating social and emotional behaviors in the mPFC (*Li et al., 2016*). Therefore, we tested the hypothesis that microinjection of OT into the PrL restores the anxiety-like behavior and social preference that were altered by preweaning PD. We implanted bilateral injection

cannulas into the PrL. After three days of recovery, subjects received an intra-PrL injection (*Figure 6A–D*) of cerebrospinal fluid (CSF), OT, OT plus oxytocin receptor antagonist (OTA) or OT plus V1aRA. The levels of anxiety-like behavior and social preference were then measured.

One-way ANOVA showed that microinjection of OT into the PrL increased the percentage of time spent in the central area by males (p<0.05) and females (1 ng: p<0.05 and 10 ng: p<0.05) that had been exposed to preweaning PD, whereas OT plus either dose of OTA had no effect (except for males injected with 10 ng OT/10 ng OTA, p<0.05). In addition, male subjects receiving 10 ng OT and 10 ng V1aRA also spent an increased percentage of time in the central area (p<0.05), whereas this treatment made no difference for females (*Figure 6E and H*, *Figure 6—source data 1*). No differences were found between treatment groups for the total distance traveled (*Figure 6F and I*, *Figure 6—source data 1*).

Two-way ANOVA revealed a main effect of the absence or presence of a stimulus mandarin vole in both sexes (male – $F_{(1,58)} = 18.135$, p<0.01; female: – $F_{(1,56)} = 27.658$, p<0.01). However, there were no interactions between the treatments and absence/presence of the stimulus vole in either sex. OT-treated males (10 ng) and females (1 ng and 10 ng) demonstrated a preference for the social stimulus over the object stimulus (p<0.008), whereas subjects treated with OT plus either dose of OTA showed no preference (except for males receiving 10 ng OT/10 ng OTA, p<0.008). In addition, microinjection of 10 ng OT/10 ng V1aRA in the mPFC still led to a preference for the social stimulus over the object stimulus in females (p<0.008). However, males that received an intra-mPFC injection of 10 ng OT/10 ng V1aRA did not demonstrate this preference (*Figure 6G and J*, *Figure 6—source data 1*).

## Optogenetic activation of oxytocinergic fibers improved social preference and decreased anxiety-like behavior in the PrL

Retrograde tracers (cholera toxin subunit B [CTB]) were injected into the PrL to visualize PVN OT neurons that project to PrL (*Figure 7—figure supplement 1*). To further explore whether activating PVN OT neuron terminals in the PrL improves social preference and emotion, we injected rAAV-Ef1α-DIO-ChR2-mCherry plus rAAV-Oxytocin-Cre or rAAV-Ef1α-DIO-mCherry plus rAAV-Oxytocin-Cre into the right PVN (*Figure 7A*). Double labeling experiments confirmed that Cre expression recapitulated the endogenous expression of OT (*Figure 7C and D*). The majority of neurons expressing ChR2-mCherry coexpressed OT neurons (male – 132/218 = 60.6%, from two voles; female – 190/268 = 70.9%, from two voles). Colocalization of AVP/ChR2-mCherry was not found in the PVN neurons (*Figure 7—figure supplement 2*). The majority of neurons that only expressed mCherry with omission of ChR2 overlapped with the OT stain (male – 147/234 = 62.8%, from two voles; female – 196/273 = 71.8%, from two voles) (*Figure 7—figure supplement 3*). Virally labeled axons were observed in the PrL, corroborating the results of the anterograde mapping experiments (*Figure 7E and F*). We implanted an optic fiber into the right PrL (*Figure 7B*). The PVN OT neuron terminals in the PrL reliably responded to pulses of 473 nm light. Furthermore, analysis of the expression of the neuronal activity marker c-fos confirmed the photostimulation-dependent activation of neurons in the PrL (*Figure 7G*). Both male (*Figure 7H*, *Figure 7—source data 1*) and female (*Figure 7I*, *Figure 7—source data 1*) ChR2-expressing voles showed significantly increased c-fos expression in the PrL when compared to control subjects (p<0.01). Optical activation of PD voles expressing ChR2-mCherry increased social preference (*Figure 7J and K*, *Figure 7—source data 1*). Control male (*Figure 7J*, *Figure 7—source data 1*) or female (*Figure 7K*, *Figure 7—source data 1*) voles showed no social preference. However, a main effect of the absence/presence of a stimulus mandarin vole was found ($F_{(1,20)} = 13.350$, p<0.01), such that optogenetically activated PVN OT neuron terminals in the PrL of ChR2-expressing PD males induced more investigations of the social stimulus than of the object stimulus (p<0.025). In addition, the treatment produced a main effect in males ($F_{(1,20)} = 5.785$, p<0.05). There was no significant interaction in males ($F_{(1,20)} = 2.011$, p=0.172). Two-way ANOVA showed a significant effect of the interaction between treatment and the absence/presence of a stimulus mandarin vole on the percentages of time in females ($F_{(1,20)} = 6.771$, p<0.05). ChR2-expressing voles engaged in more investigations of the social stimulus than the object stimulus in males (p<0.025). Meanwhile, ChR2$^+$ voles also spent more time than ChR2$^-$ voles exploring young voles (p<0.025).

On the open field test (OFT), optogenetically activated PVN-PrL OT projections improved the emotions of ChR2-expressing voles (male: $t_{(10)} = 4.452$, p<0.01, *Figure 7L*, *Figure 7—source data*

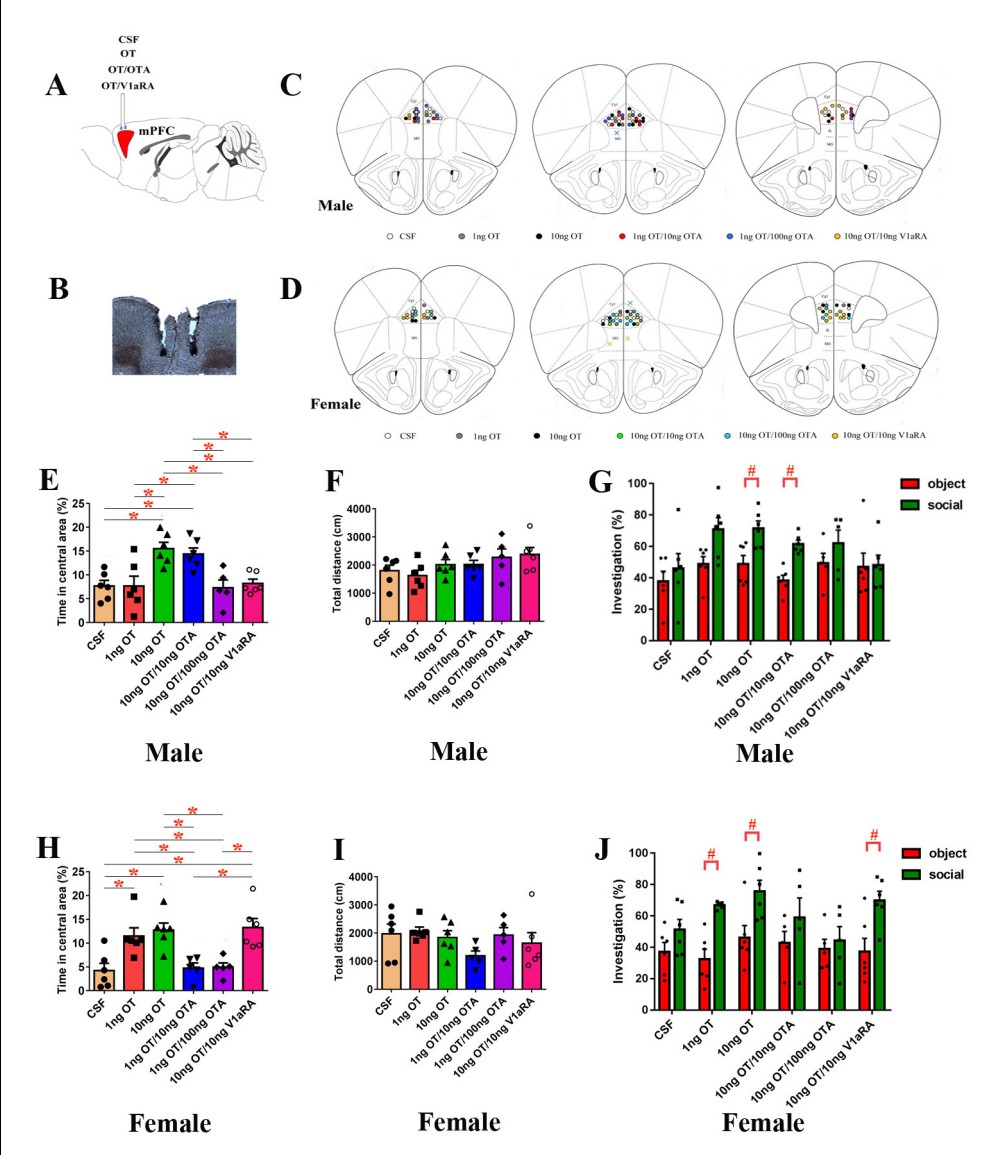

**Figure 6.** Effects of PrL OT administration on anxiety-like behavior and social preference in paternal deprivation mandarin voles. (A) Experimental schematics. (B) Histological representations of the microinjection site and (C, D) schematic diagrams showing the location of injector tips in the PrL. ×: missed. OT in the PrL is anxiolytic in both of sexes. (E, H) Percentage of time in the central area and (F, I) total distance in the open field test. One-way ANOVA. *p<0.05. OT in the PrL promotes a social preference in (G) males and (J) females. #p<0.0083 vs object stimulus. Two-way ANOVA (factors: treatment × sex). (Male: CSF – n = 6; 1 ng OT – n = 6; 10 ng OT – n = 6; 10 ng OT/10 ng OTA – n = 6; 10 ng OT/100 ng OTA – n = 5; 10 ng OT/10 ng V1aRA – n = 6. Female: CSF – n = 6; 1 ng OT – n = 6; 10 ng OT – n = 6; 1 ng OT/10 ng OTA – n = 5; 1 ng OT/100 ng OTA – n = 5; 10 ng OT/10 ng V1aRA – n = 6).

DOI: https://doi.org/10.7554/eLife.44026.014

The following source data is available for figure 6:

**Source data 1.** The percentage of time spent in the central area, the total distance in the open field, and the percentage of time spent investigating the social stimulus and the object stimulus after administration of CSF, OT, OT/OTA, or OT/V1aRA to the PrL.

DOI: https://doi.org/10.7554/eLife.44026.015

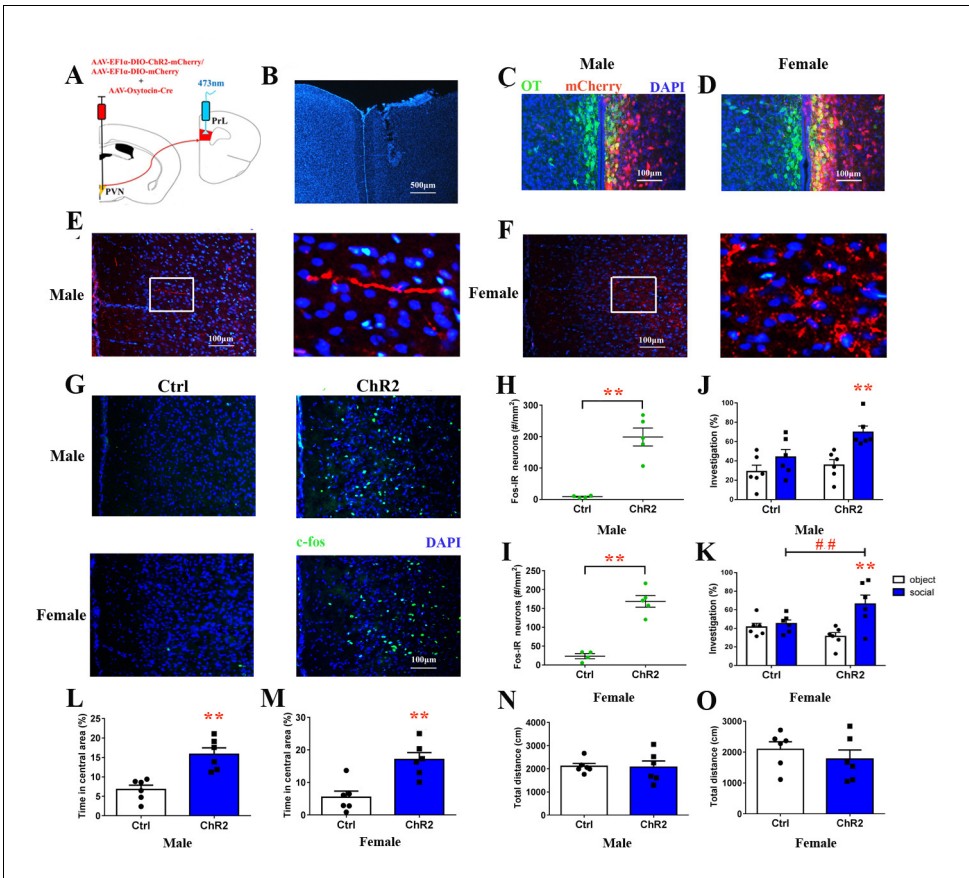

**Figure 7.** Optogenetic stimulation of OT terminals rescued changes in emotion and social preference behavior that were induced by paternal deprivation. (**A**) Schematic drawing of the locations of rAAV-Ef1α-DIO-ChR2-mCherry plus rAAV-Oxytocin-Cre or rAAV-Ef1α-DIO-mCherry plus rAAV-Oxytocin-Cre injection into the right PVN and optic fiber implants. (**B**) Immunohistological image showing the targetting of fiber implants in the right PrL. Colocalization of ChR2-mCherry (red), OT neurons (green) and DAPI (blue) in the PVN of (**C**) males and (**D**) females. Confocal images of axonal mCherry signal in the PrL of (**E**) males and (**F**) females. (**G**) Images show expression of c-fos in the PrL after photostimulation. Quantification of c-fos in the PrL of (**H**) males (Ctrl: n = 4; ChR2: n = 5) and (**I**) females (Ctrl: n = 4; ChR2: n = 5) after photostimulation. Optogenetic activation of oxytocinergic fibers in the PrL increases social preference of (**J**) males (n = 6) and (**K**) females (n = 6). **$p < 0.01$ vs. object stimulus. ## $p < 0.01$ vs. Ctrl. Two-way ANOVA (factors: photostimulation treatment × stimulus type). Activation of PVN-PrL oxytocinergic projection significantly increased the percentage of time in the central area for both (**L**) males (n = 6) and (**M**) females (n = 6), but did not influence total distance traveled for (**N**) males (n = 6) or (**O**) females (n = 6). *$p < 0.05$; **$p < 0.01$. Independent sample t-tests. Error bars indicate SEM.

DOI: https://doi.org/10.7554/eLife.44026.016

The following source data and figure supplements are available for figure 7:

**Source data 1.** Statistical result of levels of c-fos in the PrL after optogenetic activation of PrL-projecting PVN OTergic neurons during the open field, the social preference test.

DOI: https://doi.org/10.7554/eLife.44026.021

**Figure supplement 1.** Histology and immunostaining.

DOI: https://doi.org/10.7554/eLife.44026.017

**Figure supplement 2.** Immunostaining showing colocalization of mCherry (red) with OT (green) but not AVP (blue) in PVN neurons of (**A**) male and (**B**) female voles.

DOI: https://doi.org/10.7554/eLife.44026.018

**Figure supplement 3.** Representative immunostaining allowing the vizualization of (**A**) OT expression (green), (**B**) mCherry (red) and (**C**) DAPI (blue) in PVN neurons of PD voles.

DOI: https://doi.org/10.7554/eLife.44026.019

**Figure supplement 4.** After activation of PVN-to-PrL OT terminals does not elicit long-lasting effects on anxiety-like behavior and social preference in PD voles (n = 6).

*Figure 7 continued on next page*

*Figure 7 continued*

DOI: https://doi.org/10.7554/eLife.44026.020

*1*; female: t(10) = 4.051, p<0.01, *Figure 7M*, *Figure 7—source data 1*). However, the total distance covered did not differ between these two groups (*Figure 7N and O*, *Figure 7—source data 1*). The behavioral changes disappeared 8 hr after photostimulation in PD voles (*Figure 7—figure supplement 4*).

## Optogenetic inhibition of the PVN (OTergic)-PrL pathway mimics the effects of preweaning PD

Naïve voles were injected bilaterally with rAAV-Oxytocin-Cre and rAAV-Ef1α-DIO-eNpHR3.0-mCherry or rAAV-Ef1α-DIO-mCherry in the PVN (*Figure 8A*). We also found that approximately 70.1% of eNpHR3.0-mCherry$^+$ neurons in males (471/612, from two voles) or 77.6% of eNpHR3.0-mCherry$^+$ neurons in females (594/765, from two voles) overlapped with OT neurons (*Figure 8C and D*). The majority of neurons that expressed inhibitory opsin eNpHR3.0 overlapped with OT neurons (male: 427/602 = 70.9%, from two voles; female: 671/803 = 83.6%, from two voles) (*Figure 8—figure supplement 1*). eNpHR3.0-mCherry labeling showed that PVN OT neurons project their axons to the PrL (*Figure 8E and F*). An optic fiber was implanted into the bilateral PrL (*Figure 8B*). To activate eNpHR3.0, we delivered yellow light (593 nm) to the PrL. A histological analysis showed that eNpHR3.0-expressing voles had significantly less c-fos expression than mCherry$^+$ subjects in the PrL of males (p<0.01) (*Figure 8H*, *Figure 8—source data 1*) and females (p<0.01) (*Figure 8I*, *Figure 8—source data 1*). The was no significant interaction between treatment and stimulus (absence/presence) on the percentage of investigation time for males (F(1,20)=1.946, p=0.178) or females (F(1,20)=3.026, p=0.097). Both male and female control voles engaged in more investigations of the social stimulus than the object stimulus (males: p=0.017; females: p<0.01). By contrast, terminal inhibition of OT neurons projecting from the PVN to the PrL attenuated social preference in males (*Figure 8J*, *Figure 8—source data 1*) and females (*Figure 8K*, *Figure 8—source data 1*). Optogenetic inhibition of PVN (OTergic) to the PrL pathway significantly increased anxiety-like behavior (males: t(10) = 3.051, p<0.05, *Figure 8L*, *Figure 8—source data 1*; females: t(10) = 5.875, p<0.01, *Figure 8M*, *Figure 8—source data 1*). However, the total distance covered by the mCherry group was not different from that covered by the NpHR3.0-mCherry group in the open field (*Figure 8N and O*, *Figure 8—source data 1*). The naïve voles still exhibited changes in anxiety-like behavior and decreased social preference (except males) 8 hr after yellow light illumination, but the effect of yellow light stimulation disappeared within 8–24 hr (*Figure 8—figure supplement 2*).

## Discussion

The present study found that preweaning PD in mandarin vole pups increased the levels of anxiety-like behavior and attenuated social preference in male and female adults, possibly via disruption of emotional attachment. PD decreased the neuronal activity in the PrL and NAc shells and reduced the percentage of c-fos/OT-labeled cells in the PVN during the social approach test. PD also reduced the number of PVN OT-IR neurons and decreased the amounts of mPFC OTR protein in females and the OTR and V1aR proteins in males. We then demonstrated that an intra-PrL OT injection restored the anxiety-like behavior and social preference altered by preweaning PD. For the first time, we showed that optogenetic activation of the PVN-PrL OT terminals reduced anxiety-like behavior and increased social-preference behavior in PD mandarin voles. Optogenetic inhibition of PrL-projecting PVN OTergic neurons elicited anxiety-like behavior and decreased social preference in naive mandarin voles.

The finding that PND 14–21 PD increased anxiety-like behavior and reduced social preference in both sexes is consistent with studies in humans and other animals that showed that early severe deprivation is associated with behavioral abnormalities (*Rosenblum and Harlow, 1963*; *Rutter et al., 2001*). The current result is also supported by one of our previous studies that indicated that PND 14–21 offspring show emotional attachment to their fathers (*He et al., 2017*). Thus, PND 14–21 PD should disrupt emotional attachment to fathers and should adversely affect emotion and sociability. These effects are similar to previous findings that indicated that neonatal PD during PND 1–21

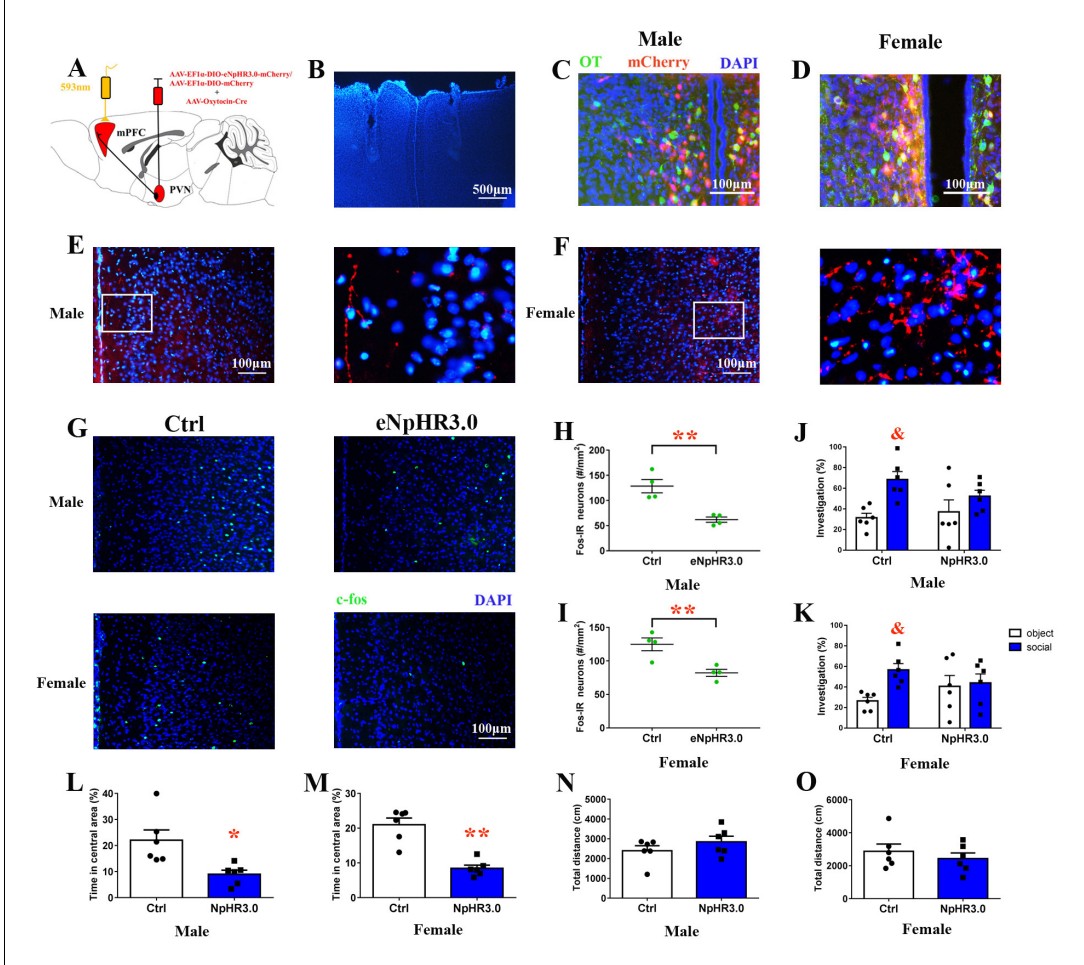

**Figure 8.** Optogenetic inhibition of OT terminals elicits anxiety-like behavior and attenuated social preference in naive voles. (A) Schematic drawing of the locations of rAAV-Ef1α-DIO-eNpHR3.0-mCherry plus rAAV-Oxytocin-Cre or rAAV-Ef1α-DIO-mCherry plus rAAV-Oxytocin-Cre injection bilaterally into the PVN and optic fiber implants. (B) Immunohistological image showing target of fiber implants in the bilateral PrL. Colocalization of eNpHR3.0-mCherry (red), OT neurons (green) and DAPI (blue) in the PVN of (C) males and (D) females. Confocal images of axonal mCherry signal in the PrL of (E) males and (F) females. (G) Images showing the expression of c-fos in the PrL after yellow light photostimulation. Quantification of c-fos in the PrL of (H) males (n = 4) and (I) females (n = 4) after photostimulation. **p<0.01. Independent sample t-tests. Optogenetic inhibition of oxytocinergic fibers in the PrL decreases the social preference of (J) males (n = 6) and (K) females (n = 6); p<0.025. Two-way ANOVA (factors: optogenetic inhibition treatment × stimulus type). Inhibition of PVN-PrL oxytocinergic projection significantly decreases the percentage of time spent in the central area for both (L) males (n = 6) and (M) females (n = 6), but did not influence total distance traveled by (N) males (n = 6) or (O) females (n = 6). **p<0.01. Independent sample t-tests. Error bars indicate SEM.

DOI: https://doi.org/10.7554/eLife.44026.022

The following source data and figure supplements are available for figure 8:

**Source data 1.** Results of statistical analysis of the levels of c-fos in the PrL after optogenetic inhibition of PrL-projecting PVN OTergic neurons during the open field and the social preference test.

DOI: https://doi.org/10.7554/eLife.44026.025

**Figure supplement 1.** Representative immunostaining of (A) OT expression (green), (B) mCherry (red) and (C) DAPI (blue) in PVN neurons of naïve voles.

DOI: https://doi.org/10.7554/eLife.44026.023

**Figure supplement 2.** After inhibition of PVN-to-PrL OT terminals does not elicit long-lasting effects on anxiety-like behavior and social preference in naïve voles (n = 6).

DOI: https://doi.org/10.7554/eLife.44026.024

increases anxiety (*Yu et al., 2011*) and reduces sociability (*Bambico et al., 2015*; *Farrell et al., 2016*). We infer that the disruption of attachment between pups and fathers during PND 14–21 increases the level of anxiety and reduces the level of sociability.

The present study found that PND 14–21 PD reduced mPFC OTR protein in both sexes but did not affect AVP levels. The mPFC may receive long-range axonal projections from OT-producing neurons (*Knobloch et al., 2012*). A previous study found that OT in the PrL region of the mPFC decreased anxiety regardless of sex and that neither AVP nor OTR-A affected anxiety-like behavior (*Sabihi et al., 2014b*). Blocking OTR in the PrL enhances postpartum anxiety but has no effect on anxiety in virgin females (*Sabihi et al., 2014a*). OTR knockout mice display deficits in social-approach behavior (*Nishimori et al., 2008*) and social memory (*Lee et al., 2008*). Thus, OTR in the mPFC may play different roles under different physiological and pathological conditions. PVN OT neurons predominantly express vesicular glutamate transporter 2, suggesting that depolarization of these neurons is coupled with synaptic glutamate release in their projections, such as those to the mPFC (*Kawasaki et al., 2005*; *Johnson and Young, 2017*). In addition, a specific class of interneurons (OxtrINs) in the mPFC is critical for the modulation of social and emotional behaviors in both sexes (*Li et al., 2016*). It is likely that the release of glutamate by OT neurons activates local GABA-interneurons in the mPFC, leading to reduced anxiety and regulated social behavior. Thus, we conclude that disruption of the attachment between pups and fathers increases the levels of anxiety and impairs social preference, possibly by decreasing the OTR protein in the mPFC.

PND 14–21 PD decreased the V1aR protein in males but had no effect on the level of the V1aR protein in females. Sex differences in the OT system may, therefore, be implicated in sex-specific regulation of impaired social behavior (*Smeltzer et al., 2006*; *Dumais and Veenema, 2016*). V1aR and OTR show distinct and largely nonoverlapping expression in the rodent brain (*Dumais and Veenema, 2016*). Female prairie voles have greater densities of OTR binding but lower densities of V1aR binding than males in the mPFC (*Smeltzer et al., 2006*). Intracerebroventricular (ICV) injections of OT induce social communication by activating V1aR in male Syrian hamsters (*Song et al., 2014*). Studies have shown that some of the prosocial effects of OT may be mediated by V1aR in males (*Sala et al., 2011*; *Ramos et al., 2013*) and that V1aR knockout mice have impaired social interactions (*Egashira et al., 2007*). V1aR is a G protein-coupled receptor, similar to OTR. We speculate that OT might regulate anxiety-like behavior and social preference in males via OTR and V1aR in the PrL.

PND 14–21 PD did not affect the density of OTR-IR in either sex, which isinconsistent with the previous finding that PD (PND 1–21) alters the level of OTR mRNA in the NAc (*Cao et al., 2014*). These discrepancies may result from different deprivation periods. Nevertheless, this suggestion requires further study.

A major finding of the current study is that microinjection of OT directly into the PrL of males (10 ng) and females (1 ng and 10 ng) restored the changes to anxiety-like behavior and social preference caused by PD, whereas voles treated with OT plus either dose of OTA did not exhibit a reversal of any kind (except for the male 10 ng OT/10 ng OTA group). For the first time, we found that stimulation of endogenous OT fibers by optogenetic in the PrL also rescued emotion and social-preference behavior. However, optogenetic inhibition of endogenous OT release was anxiogenic and decreased social preference in naive voles. This result is consistent with previous findings that OT in the PrL reduces anxiety-like behavior in both sexes (*Sabihi et al., 2014b*). Injection of highly specific OTA into the PrL region of the mPFC increases anxiety-like behavior in postpartum females (*Sabihi et al., 2014a*), and OTR blockade in the postpartum PrL impairs maternal care behavior and enhances maternal aggression (*Sabihi et al., 2014a*). OxtrINs have been identified in the mouse mPFC that express OTR and were shown to be activated in response to OT (*Nakajima et al., 2014*). OxtrINs are important for the modulation of social and emotional behaviors in males and females and are involved in a molecular mechanism that acts on local mPFC circuits to coordinate responses to OT and corticotropin-releasing hormone (*Li et al., 2016*). Similarly, OT infusion into the PrL reverses amphetamine-induced deficits in social bonding (*Young et al., 2014*). OT and dopamine (DA) interactions regulate affiliative social behaviors (*Liu and Wang, 2003*; *Shahrokh et al., 2010*). Absence of the father in the monogamous California mouse impairs social behavior and decreases pyramidal neuronal responses to DA in the mPFC (*Bambico et al., 2015*). Injection of oxytocin into the ventral tegmental area increases the extracellular DA concentration in the dialysate from the mPFC (*Sanna et al., 2012*). Therefore, there is likely that there is a reciprocal interaction between

OT and DA that regulates social behavior in the mPFC, and that mPFC OT may be a promising target for treating emotional and social disorders that are induced by adverse early experiences.

Changes in OT pathways that are induced by severe early life stress can be transmitted to the next generation (*Toepfer et al., 2017*). Dysfunction of OT system in adulthood caused by early life stress can reduce affiliated contact with offspring and consequently can result in changes in the OT systems of the offspring (*Rilling and Young, 2014*). In the present study, preweaning paternal deprivation reduced OT neurons in the PVN, reduced levels of OTR in the mPFC and affected relevant behaviors. Consistent with previous reports, PD from weaning (*Feng et al., 2019*) to adolescence (*Wang et al., 2017*) decreased the number of OT-positive neurons in the PVN of mandarin voles. Thus, parental deprivation impacts the oxytocin system of the offspring on a permanent basis. We also found that disruptions of early emotional attachment reduced the number of OT-IR neurons in the PVN of males and females. Maternal separation is known to decrease the number of PVN OT-IR neurons in females but not in males (*Veenema et al., 2007*). These discrepancies may result from the different treatments (maternal separation and PD) and different sensitivities to neonatal stress in different species. Furthermore, we showed that activation of the PVN-PrL OTergic projection improved anxiety-like behavior and social preference. Previous studies found that social approach (*Peñagarikano et al., 2015*) and mood (*Grund et al., 2017*) were improved by activating endogenous oxytocin neurons in the PVN with chemogenetics. We speculate that these changes were also partly due to activation of the PVN-PrL OT pathway.

Mandarin voles can be used as an animal model to investigate the effects of early emotional attachment disruption on the adult brain, on behavior and on the underlying mechanism (*He et al., 2017*). Disruption of the early emotional attachment between pups and fathers impairs emotional and social behaviors and leads to OT system dysfunction in the brain. We provide intriguing evidence that the site-specific action of OT in the PrL projected from the PVN has potential beneficial effects on the recovery of the emotional and social dysfunction induced by the disruption of early emotional attachment. The modulation of OT on emotion and social behaviors was sex-specific. Therefore, OT can be potentially targeted to ameliorate the social and emotional deficits resulting from early adverse experiences.

## Materials and methods

**Key resources table**

| Reagent type (species) or resource | Designation | Source or reference | Identifiers | Additional information |
|---|---|---|---|---|
| Antibody | anti-oxytocin mouse monoclonal antibody | Millipore | Cat#MAB5296 RRID:AB_2157626 | concentration: 1:7500 |
| Antibody | anti-vasopressin rabbit polyclonal antibody | Millipore | Cat#AB1565 RRID:AB_90782 | concentration: 1:4000 |
| Antibody | anti-c-fos rabbit polyclonal antibody | Abcam | Cat#ab190289 RRID:AB_2737414 | concentration: 1:1500 |
| Antibody | anti-oxytocin receptor rabbit monoclonal antibody | Abcam | Cat#ab181077 | concentration: 1:2000 |
| Antibody | anti-vasopressin receptor 1A (V1aR) goat polyclonal antibody | GeneTex | Cat#GTX89114 RRID:AB_10724608 | concentration: 1:7000 |
| Antibody | anti-β-tubulin mouse monoclonal antibody | ComWin Biotechnology | Cat#CW0098M | concentration: 1:5000 |
| Antibody | anti-rabbit goat conjugated with TRITC | Jackson ImmunoResearch | Cat#111-025-003 RRID:AB_2337926 | concentration: 1:200 |

*Continued on next page*

*Continued*

| Reagent type (species) or resource | Designation | Source or reference | Identifiers | Additional information |
|---|---|---|---|---|
| Antibody | anti-rabbit goat conjugated with DyLight 405 | Jackson Immunoresearch | Cat#111-475-003 RRID:AB_2338035 | concentration: 1:200 |
| Antibody | anti-rabbit goat conjugated with DyLight 488 | Boster | Cat#BA1127 | concentration: 1:200 |
| Antibody | anti-mouse goat antibody conjugated with DyLight 488 | Boster | Cat#BA1126 | concentration: 1:200 |
| Antibody | anti-rabbit goat conjugated with horseradish peroxidase | ZhongShan Goldenbridge | ZB-2301 RRID:AB_2747412 | concentration: 1:10,000 |
| Antibody | anti-mouse goat conjugated with horseradish peroxidase | ZhongShan Goldenbridge | ZB-2305 RRID:AB_2747415 | concentration: 1:10,000 |
| Antibody | anti-goat rabbit conjugated with horseradish peroxidase | ZhongShan Goldenbridge | ZB-2306 | concentration: 1:10,000 |
| Reagent | normal goat serum | Boster | Cat#AR0009 | |
| Virus | AAV-Ef1α-DIO-ChR2-mCherry | BrainVTA | Cat#PT-0002 | |
| Virus | AAV-Ef1α-DIO-eNpHR3.0-mCherry | BrainVTA | Cat#PT-0007 | |
| Virus | AAV-Ef1α-DIO-mCherry | BrainVTA | Cat#PT-0013 | |
| Virus | AAV-Oxytocin-Cre | BrainVTA | Cat#PT-0263 | |
| Chemical compound, drug | radioimmunoprecipitation assay buffer (RIPA) | Solarbio | Cat#R0010 | |
| Chemical compound, drug | enhanced chemiluminescence (ECL) reagent | Millipore | Cat#WBKLS0500 | |
| Chemical compound, drug | antifade solution | Boster | Cat#AR1109 | |
| Chemical compound, drug | 4′,6-diamidine-2′-phenylindole dihydrochloride (DAPI) | Boster | Cat#AR1176 | |
| Chemical compound, drug | Cholera Toxin Subunit B (CTB)—594 | Thermo Fisher Scientific | Cat#C34777 | |
| Software, algorithm | OBSERVER v5.0 | Noldus | https://www.noldus.com/knowledge-base/observer-50 | |
| Software, algorithm | SPSS | IBM | RRID:SCR_002865 | |
| Software, algorithm | ImageJ | NIH | RRID:SCR_003070 | |
| Software, algorithm | GraphPad Prism 5 | GraphPad | RRID:SCR_002798 | |
| Software, algorithm | SuperMaze | Shanghai XinRuan | XR-XJ117 | |
| Software, algorithm | SocialScan | Clever Sys | http://cleversysinc.com/CleverSysInc/home/software/socialscan/ | |
| Other | PVDF membranes | Millipore | C3117 | |

## Subjects

Animals were a laboratory-reared offspring originating from a wild population of mandarin voles in Henan, China. Animals were maintained on a 12:12 light:dark cycle with unlimited access to food (carrot and rabbit chow) and were provided with water and cotton nesting material in polycarbonate cages (44 cm x 22 cm x 16 cm). All procedures were approved by the Animal Care and Use Committee of Shaanxi Normal University and were in accordance with the Guide for the Care and Use of Laboratory Animals of China.

For the paternal deprivation (PD) treatment, fathers ($F_1$ generation) were removed permanently from the home cage after pups ($F_2$ generation) were 14 days of age until weaning at PND 21. For the biparental care control group (PC), all family members were housed in their home cage and left undisturbed until pups were weaned at 21 days of age. Offspring at 70 days of age were tested using the behavioral paradigms below. For females, only experimental data from diestrous individuals were included to avoid effects from the estrous cycle.

## Open field test

$F_2$ mandarin voles were observed in the open field. Subjects were placed in a center of an open-field arena (50 cm x 50 cm x 25 cm), and the duration and distance moved within the center or periphery was recorded using an automated system (SocialScan 2.0, Clever Sys, Reston, VA, USA). Measures include the proportion of time spent in the central area and total distance moved in the open field test (OFT).

## Light and dark test

The light and dark apparatus was divided into two chambers: white compartment (2/3) and dark compartment (1/3). Voles were released into the center of the light compartment and have 5 min to explore the arena. Videotracking equipment (SuperMaze) was used for the voles' duration in the light arena.

## Social-preference test

Immediately after the open field test, the social preference test (SPT) was carried out. The SPT was based on the social approach-avoidance paradigm described previously (*Qiao et al., 2014*). Briefly, prior to testing, voles were placed in a box. The test box (50 cm length x 50 cm width x 24 cm height) was constructed of white glacial polyvinylchloride. After 5 min of habituation, an empty wire-mesh cage (object stimulus; 10 cm length x 10 cm width) was placed near one side wall of the arena for 10 min, which was then exchanged for a cage containing an stranger same-sex con-specific (social stimulus) for an additional 10 min. Behavioral responses to an empty cage or to a cage with a stimulus individual were videotaped, and scored and quantified afterwards using OBSERVER v5.0 (vNoldus, NL). Measures included the time spent investigating the object and social stimulus. Data are presented as investigation time/total time (10 min) x 100%.

## Social approach test

The social approach test was a modified version of that described previously (*Gunaydin et al., 2014*; *Hung et al., 2017*). After 5 min of habituation, a stranger same-sex conspecific juvenile vole (3–5 weeks) or novel object (a Rubik's cube) was then placed in the home cage for 10 min. The 5-cm-wide area in front of the wire-mesh cage and the 8-cm-wide area in the left and wire-mesh cage were considered to be zones of olfactory, visual and acoustic contact between subjects and stimulus voles. Duration of olfactory investigation was also recorded and analyzed in real time using an automated system (SuperMaze).

## Immunohistochemistry

Experimental mandarin voles were anesthetized with pentobarbital sodium. Brains were removed from the skull and placed in 4% paraformaldehyde for 3 d. Brains were then dehydrated in 30% sucrose solution until saturated at 4°C.

Thick coronal slices were prepared on a cryostat (CM1950, Leica, Germany). Sections were rinsed with 0.01 M phosphate buffer solution (PBS) for 10 min following incubation with 0.6% $H_2O_2$ for 20 min, and rinsed for 3 × 5 min with 0.01 M PBS. Sections were then preincubated for 60 min with

normal goat serum. Sections were incubated for 48 hr at 4°C in OT, c-fos and AVP diluted in anti-body diluent. Next day, sections were rinsed in 0.01 M PBS for 3 x 5 min and incubated with the secondary antibody for 60 min in a 37°C water bath. The nuclei were stained by DAPI dye for 10 min. Afterwards, sections were rinsed for $3 \times 5$ min with 0.01 M PBS and fixed with antifade solution.

Image acquisition was performed with a microscope and Nikon camera. For each vole, the number of OT-IR neurons (PVN) were counted, three representative sections from anterior to posterior that were anatomically matched between subjects were chosen to minimize variability. The count of OT-IR cells (*Song et al., 2010*) followed the method described in the previous report. Six brain sections for each vole were counted to quantify the c-fos-positive cells and the OT/c-fos-positive cells. All immunohistochemistry procedures were also performed on negative-control sections (the primary antibody was not added). An observer blind to experimental conditions performed the entire analysis. The number of OT-positive neurons were quantified from images acquired under a 10X objective using the cell counter; for c-fos-positive cells and OT/c-fos-positive cells, image acquisition was performed under a 20X objective; these were averaged across three nonoverlapping sections in an evenly spaced series for each animal.

## Western blotting

Voles from the PC and PD groups were sacrificed by rapid decapitation at 70 days of age. Brains were immediately extracted and frozen in dry ice. Coronal sections (200 μm) were cut on a cryostat and frost mounted onto microscope slides. Bilateral tissue punches with a 1 mm diameter were taken from the entire mPFC (Cg and PrL), NAc and PVN and stored at −80°C until processing. Total proteins were extracted with RIPA lysis buffer containing protease inhibitors. Protein samples (20 μg/lane) were separated by sodium dodecyl sulphate-polyacrylamide gel electrophoresis and transferred to PVDF membranes. The membrane was incubated with the following diluted primary antibodies: OTR, V1aR, AVP and β-tubulin, at 4°C overnight. Following washing, the membrane was incubated with horseradish-peroxidase-conjugated secondary antibodies; membranes were revealed with ECL and exposed on Luminescent Imaging (Tanon 6200 Luminescent Imaging Workstation, Tanon). Quantification was performed using ImageJ software, and all signals were normalized within the same membrane to β-tubulin.

## Stereotaxic cannulation and microinjection

At 70 days of age, mandarin voles in the PD group were anesthetized by isoflurane, and 26-gauge bilateral steel guide cannulae (R.W.D. Life Science, Shenzhen, China) were stereotaxically implanted aimed at the PrL (coordinates from bregma: anterior, 2.2 mm; bilateral, ±0.5 mm; ventral, 2.2 mm). Voles were allowed 3 d of post-operative recovery. One of three solutions — CSF (200 nl/side), CSF containing OT (1 ng/200 nl/side or 10 ng/200 nl/side), or CSF containing OT (male 10 ng OT; female: 1 ng OT) plus OTA (10 ng/200 nl/side or 100 ng/200 nl/side) — was injected into the PrL. All microinjections were made with a 33-gauge needle that extended 1 mm below the guide cannula and were delivered at a rate of 0.1 μl/min as previously described (*Young et al., 2014*). Data were excluded if the tips were located in other brain regions. The final size of each group was six except for the 10 ng OT/100 ng OTA male group which numbered five, and the 1 ng OT/10 ng OTA and 1 ng OT/100 ng OTA female groups (both n = 5).

## Optogenetics and behavioral tests

At 28 days of age, mandarin voles were anesthetized with 0.2% pentobarbital sodium and placed in a stereotaxic apparatus (R.W.D. Life Science, Shenzhen, China). After craniotomy, the PD voles were injected (40 nl /min) with rAAV-Ef1α-DIO-ChR2-mCherry: rAAV-Oxytocin-Cre (1:1, total 300 nl) or rAAV-Ef1α-DIO-mCherry: rAAV-Oxytocin-Cre (1:1, total 300 nl) into the right PVN (coordinates from bregma in mm: −0.7 posterior, 0.1 lateral, −5.4 ventral); the naive voles were injected (40 nl /min) with rAAV-Ef1α-DIO-eNpHR3.0-mCherry: rAAV-Oxytocin-Cre (1:1, total 300 nl) or rAAV-Ef1α-DIO-mCherry: rAAV-Oxytocin-Cre (1:1, total 300 nl) into the bilateral PVN (coordinates from bregma: −0.7 mm posterior, ±0.1 mm lateral, −5.4 mm ventral). AAV titers were about $10^{12}$ gc/ml. All injections were kept in place for an additional 10 min to allow the virus diffusion from the injection needle. Five weeks after virus injection, voles were implanted with optical fibre (OD = 250 μm, 0.37 NA; Shanghai Fiblaser) targeted to the right PrL (coordinates from bregma: anterior, 2.2 mm;

lateral, −0.3 mm; ventral, 1.8 mm) or the bilateral PrL (coordinates from bregma: anterior, 2.2 mm; lateral, ±0.32 mm with a 5° angle; ventral, 1.81 mm). Optogenetic behavioral tests started 6 weeks after virus injection.

For the activation or inhibition of oxytocinergic terminals in the PrL, optogenetic parameters (ChR2: 10 ms, 30 Hz, 8 s on and 2 s off cycle, 15 ~ 20 mWatts; eNpHR3.0: 5 ms, 30 Hz, 8 s on and 2 s off cycle, ~10 mWatts) were modified as in previous studies (*Marlin et al., 2015*; *Hung et al., 2017*) with a blue light laser (473 nm) or a yellow light laser (593 nm). In behavioral tests, voles were placed in the open field and allowed to explore freely for 5 min. Next, the social approach-avoidance test was carried out immediately. Subjects received total 25 min blue laser pulses in during whole test phase. After two days, to detect the c-fos expression actived by the optogenetic stimulus, voles were placed in an open field for 5 min and were sacrificed 90 min later.

## Data analysis

Parametric tests were used because all data were normally distributed according to one-sample Kolmogorov–Smirnov tests. Independent sample t-tests (two-tailed) were used to assess differences in behavior in the open field (PC vs. PD; Ctrl vs. ChR2; and Ctrl vs. eNpHR3.0), levels of OTR, V1aR and AVP protein (PC vs. PD) in different brain regions, and the number of c-fos-IR positive neurons (Ctrl vs. ChR2; Ctrl vs. eNpHR3.0). The significance level was set at $p < 0.05$. The social preference test (factors: treatment $\times$ stimulus); the social approach test (factors: treatment $\times$ stimulus); the behavioral changes following optogenetic activation (factors: treatment $\times$ stimulus); PVN OT-IR (the number of positive neurons) (factors: treatment $\times$ sex); and the number of c-fos-IR-positive neurons (factors: treatment $\times$ stimulus) were analyzed using two-way ANOVA. Paired t-tests (two-tailed) were used to compare approach/avoidance with Bonferroni correction for multiple comparisons. The open field (pharmacology and the behavioral changes following optogenetic activation) was analyzed using one-way ANOVA. Data were presented as mean + SEM, all statistical procedures were performed using SPSS 17.0.

## Acknowledgements

This work was supported by the National Natural Science Foundation of China (grants 31372213 and 31670421) and Fundamental Research Funds for Central University (GK201903065). The contribution by LJY was also supported by P50MH100023 and P510D11132. We would like to thank Larry J Young for his insightful comments on this study and extensive editing of the manuscript. We are also grateful to Xin-Ming Ma for his advice on the manuscript and discussions. Prof. Tai designed the study and wrote the protocol. Zhixiong He conducted the majority of the work and the statistical analysis, and wrote the first draft of the manuscript. Limin Wang, Wei Yuan, Laifu Li, Jing zhang, Hui Qiao and Rui Jia conducted some experiments, discussed the results and provided constructive comments. Qianqian Guo, Wenjuan Hou, Yang Yang and Luo Luo helped to care for the voles.

## Additional information

### Funding

| Funder | Grant reference number | Author |
| --- | --- | --- |
| Fundamental Research Funds for Central University of China | GK201903065 | Zhixiong He |
| National Institutes of Health | P50MH100023 | Larry Young |
| Yerkes Regional Primate Research Center | P51OD11132 | Larry Young |
| National Natural Science Foundation of China | 31372213 | Fadao Tai |
| National Natural Science Foundation of China | 31670421 | Fadao Tai |

The funders had no role in study design, data collection and interpretation, or the decision to submit the work for publication.

## Author contributions

Zhixiong He, Conceptualization, Data curation, Methodology, Writing—original draft, Writing—review and editing; Larry Young, Xin-Ming Ma, Conceptualization, Methodology, Writing—review and editing; Qianqian Guo, Luo Luo, Data curation, Investigation, Methodology; Limin Wang, Conceptualization, Investigation, Methodology; Yang Yang, Resources, Investigation, Methodology; Wei Yuan, Data curation, Software, Formal analysis, Methodology; Laifu Li, Conceptualization, Formal analysis, Investigation; Jing Zhang, Resources, Data curation, Investigation; Wenjuan Hou, Conceptualization, Resources, Investigation, Methodology; Hui Qiao, Conceptualization, Resources, Methodology; Rui Jia, Conceptualization, Supervision, Methodology; Fadao Tai, Conceptualization, Resources, Formal analysis, Supervision, Funding acquisition, Methodology, Writing—original draft, Writing—review and editing

## Author ORCIDs

Hui Qiao (iD) http://orcid.org/0000-0002-2476-4342
Fadao Tai (iD) http://orcid.org/0000-0002-6804-4179

## Ethics

Animal experimentation: Animal experimentation: All procedures were performed in strict accordance with the Guide for the Care and Use of Laboratory Animals of China. All of the animals were handled according to approved by the Animal Care and Use Committee of Shaanxi Normal University. All surgery was performed under sodium pentobarbital anesthesia, and every effort was made to minimize suffering.

## Decision letter and Author response

Decision letter https://doi.org/10.7554/eLife.44026.028
Author response https://doi.org/10.7554/eLife.44026.029

# Additional files

## Supplementary files

• Transparent reporting form
DOI: https://doi.org/10.7554/eLife.44026.026

## Data availability

All data generated or analysed during this study are included in the manuscript and supporting files. Source data files have been provided for all Figures.

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
