## [Decision Letter]

[Editors’ note: a previous version of this study was rejected after peer review, but the authors submitted for reconsideration. The first decision letter after peer review is shown below.]

Thank you for submitting your work entitled "Increased anxiety and decreased sociability in adulthood following paternal deprivation involve oxytocin in the mPFC" for consideration by *eLife*. Your article has been reviewed by three peer reviewers, and the evaluation has been overseen by a Senior/Reviewing Editor. The reviewers have opted to remain anonymous.

Our decision has been reached after consultation between the reviewers. Based on these discussions and the individual reviews below, we regret to inform you that your work will not be considered further for publication in *eLife*.

The reviewers noted that this study addresses an interesting topic and agreed that Mandarin voles represent a good model for the investigation of paternal deprivation. However, they identified a need for more experiments to support the conclusions drawn, as well as areas where the presentation and analysis of the results could be improved. Most importantly, they highlighted the limited novelty of the study and the lack of mechanistic insight provided. On the basis of these concerns, we have concluded that this study does not provide the breakthrough for the field that would justify publishing in a broad interest journal like *eLife*.

Reviewer #1:

This findings from this study show that paternal deprivation during the pre-weaning period (PD14-21) of mandarin voles leads to increased anxiety-like behavior and diminished social preferences and implicate disruption of the mPFC OT system in these effects. Overall the question addressed is interesting and the results are straightforward although I do have several comments/concerns:

1) The authors suggest that the detrimental effects of PD are due to disrupted "emotional attachment" but they provide little if any justification for this. Showing increases in anxiety behavior and attenuated social preference is not evidence that disrupted attachment is the casual factor.

2) The use of OTR receptor antibodies has been called into question. Therefore, more details should be provided about the specificity of this antibody and any control experiments that were performed. Ideally, the authors would corroborate the current data with another method.

3) In various places throughout the manuscript the authors discuss the effects of OT on the V1a receptor and even suggest in the Discussion that OT may regulate anxiety-like behavior and social preference in males via OTR and V1aR in the cortex. Thus, it seems surprising that in the pharmacology experiment (experiment 3) they didn't include an additional group that was co-administered OT with an AVP receptor antagonist.

4) The description of the statistical results for experiment one is inaccurate. Specifically, the% Investigation data for males and females are described the same way except that for males there was an interaction and for females there was only a main effect of stimulus.

5) For experiment 3, the possibility of OT diffusion beyond the PL mPFC wasn't addressed. Also, it was mentioned that data were excluded if cannula tips were located in other brain regions. I wonder if the number of subjects that had missed cannula placements is sufficient to form a "miss group" as a control for site specificity.

Reviewer #2:

This manuscript reports on the role of the oxytocin system in the anxiety produced by parental separation during the preweaning period but after the lactation period in Mandarin vole pups. The novelty of the report is the time period explored and the use of a highly social mammal in which parental bonds extend beyond the period of physical dependence. The authors find that anxiety following parental separation can be reduced by oxytocin infusions into the PLC and report a decrease in oxytocin innervation of the same area following depravation. The review of the literature is highly scholarly and the findings are appropriately placed in the context of the field.

The weakness of the report is the lack of novelty. Oxytocin is arguably the most intensely investigated peptide on the planet for its role in social bonding and anxiety. Indeed the authors make such a compelling argument that the anxiety induced by the loss of a parent must be mediated by oxytocin that any finding to the contrary would have been truly surprising, and interesting. But as it stands, the current report only modestly adds to the existing and extensive literature.

What would be an advance would be some mechanistic insight into how oxytocin innervation is impacted by early life depravation of a parent. Do the oxytocin neurons die? Or is expression of the gene repressed? Are there enduring epigenetic modifications? Etc. However before that line of inquiry were to be pursued it would also be important to provide far more convincing evidence that the innervation has in fact changed. The images provided here are not reassuring from a quantitative standpoint.

Reviewer #3:

He et al. examined the effects of paternal deprivation (PD) of Mandarin voles on anxiety and social preference of adult males and females. Such effect has been reported before and the present manuscript focuses on PD during pre-weaning period. The main findings are: 1) Voles subjected to PD at prenatal days (PND)14-21 display heightened anxiety and reduced social preference. 2) PD males and female voles have reduced OT-positive fibers in the mPFC. PD males have reduced oxytocin (OT)-positive neurons in the PVN, while females display the opposite phenotype. 3) PD also affected the levels of OT and AVP receptors (OTR and V1a) in different brain regions and the levels of serum corticosterone (CORT) and OT were altered in females. 4) injection of oxytocin into the mPFC could somewhat reverse the anxiety and social behavior phenotypes.

On a positive note, the Mandarin voles are, as mentioned in the manuscript, a good model for the investigation of PD, and the establishment of the PND 14-21 PD paradigm separates the emotional deprivation from the physiological. However, in my opinion, the contribution of this manuscript to the field is incremental to what is already known for this and other species. As mentioned by the authors, the effects of PD on both the behavioral and anatomical phenotypes reported in this manuscript were shown before. The effects of neonatal paternal deprivation or early deprivation on anxiety and social behaviors adults and juveneille Mandarin voles [Wang et al., 2012, and Jia et al., 2009]. Moreover, the reduced number of OT-IR neurons in the PVN following PD was previously shown in juvenile mandarin voles [Wang et al., 2012]; Lower serum OT was previously found in both juvenile [Wang et al., 2012] and adult PD female voles [Cao Y. et al., 2014]; Lack of significant increase in serum CORT in PD males was previously established [Yu et al., 2015].

In sum, the novelty of the present manuscript is limited to a specific paradigm of PD of Mandarin voles at PND 14-21. Moreover, the manuscript does not provide sufficient data regarding the mechanism of the behavioral and neuroanatomical deficits. The behavioral rescue with OT injection (Figure 7) is somewhat novel but very limited and preliminary in terms of identifying the exact neuronal circuit which is affected by OT administration. I also have some major technical concerns which are listed below.

Specific major concerns:

Figure 1: Additional anxiety-related behavioral assays (i.e. elevated plus maze, Dark-light transfer assay) would assist in further characterization of anxiety in PD voles, and could contribute to establishing the differential response to PD in females vs. males.

Figure 3: The images showing the effect of PD on PLC OT-IR fibers are of very poor quality and not convincing. The reader will find it hard to believe that hundreds of OT-IR fibers were reliably quantified.

Figure 4: This figure is all about a negative result (no difference in nucleus accumbens OT fibers) and it is not necessary to show as a main figure. It will be more appropriate as either a supplementary figure or as part of Figure 3.

Figure 5: The quality of the Western blots showing the levels of OTR and V1aR is unacceptable. Firstly, the gels are cropped so the specificity of the antibodies (which are known to be problematic) cannot be appreciated. Secondly, some of the protein bands are so weak and close to the background level, at some panels and the loading controls of anti-Tubulin are not equal and therefore the experimental evidences regarding the effects of PD on OTR (Figure 5A and B) and V1aR (Figure 5C) are not sufficiently supported.

Figure 6: This figure demonstrates the effects of PD on OT serum levels in females. Measuring OT serum levels using ELISA kits is criticized in the field [see: Leng and Sabatier J Neuroendocrinol. 2016 Oct;28(10). doi: 10.1111/jne.12413]. In particular, the method of extraction is critical and may cause artifacts. The authors state that this was performed using "vole-specific enzyme-linked immunosorbent assay (Shanghai Xitang Biotechnology, Shanghai, China), according to the manufacturers' instructions." however, they do not provide details or a reference regarding the validity of this assay.

Figure 7: Effects of PLC OT administration on anxiety and social preference behaviors of PD voles. The presentation of data in this figure is very confusing. In particular, it is not clear whether the authors compared the behavioral effects of OT on PD vs. naïve (non-PD) voles. This should be clarified.

[Editors’ note: what now follows is the decision letter after the authors submitted for further consideration.]

Thank you for submitting your article "Increased anxiety and decreased sociability in adulthood following paternal deprivation involve PVN-mPFC OTergic pathway" for consideration by *eLife*. Your article has been reviewed by two peer reviewers, and the evaluation has been overseen by Andrew King as the Senior and Reviewing Editor. The reviewers have opted to remain anonymous.

The reviewers have discussed the reviews with one another and the Reviewing Editor has drafted this decision to help you prepare a revised submission.

Summary:

The reviewers agreed that the additional experiments carried out have strengthened this paper, overcoming several of the issues that were raised previously, as well as providing a more direct link between the effects of parental deprivation and activity in the projection from oxytocinergic PVN neurons to the mPFC. They differed, however, in their views about the significance of these findings.

The reviewers' comments are included below. In addition, the following points have been made by the Reviewing Editor and need to be addressed in any revision:

1) The optogenetic experiments require further information and important controls are missing. These include making injections that omit ChR2 from the viral construct to demonstrate the specificity of the effect; quantifying the co-localization of ChR2-mCherry and oxytocin expression in the PVN; providing more details about the overall duration over which optogenetic stimulation was delivered. How long do the behavioral changes last following optogenetic activation? This is particularly important if you wish to argue that abnormal social behavior can be rescued in this fashion, and relates to the concern raised by reviewer 1 about the permanence (or otherwise) of the effects of parental deprivation. Can the behavioral changes be rescued by providing PD voles with increased social interactions in later life? The optogenetic experiments would also be strengthened if the behavioral consequences of expressing an inhibitory opsin in the PVN-mPFC axons were measured.

2) The quality of the writing is poor and in places does not make sense. The meaningless expression 'socially mandarin vole' is still in the Abstract. The reference list is also incomplete.

3) There are many plots in which histograms with error bars are used; individual data points should be shown rather than error bars.

Reviewer #1:

The authors have done some additional work and corrected some weaknesses but my initial concern was that the finding was predictable and so novelty is low. Understanding how parental deprivation impacts the oxytocin system on a permanent basis would be an advance. That advance is not made here.

Reviewer #2:

The authors have addressed the majority of my comments to the extent that I now in opinion that the manuscript is suitable for publication in *eLife*. In particular they have added new experiment showing to investigate the effect of pre-weaning PD on neuronal activities of PrL, NAc using anti-Fos staining as well as activity of OT neurons following exposure to a juvenile vole using double staining ot OT and c-Fos. They also added a rescue experiment showing that optogenetic activation of OT terminals in the mPFC decreased anxiety social preference and decreased anxiety-like behaviors that were induced by PD. The quality of the figure was also greatly improved.

---

## [Author Response]

[Editors’ note: the author responses to the first round of peer review follow.]

The reviewers noted that this study addresses an interesting topic and agreed that Mandarin voles represent a good model for the investigation of paternal deprivation. However, they identified a need for more experiments to support the conclusions drawn, as well as areas where the presentation and analysis of the results could be improved. Most importantly, they highlighted the limited novelty of the study and the lack of mechanistic insight provided. On the basis of these concerns, we have concluded that this study does not provide the breakthrough for the field that would justify publishing in a broad interest journal like eLife.

We have revised the manuscript according reviewers’ suggestion. Especially, to further reveal underlying mechanisms, optogenetic experiments have been added to investigate involvement of PVN OT neurons projection to PrL in effects of paternal deprivation. In addition, effects of pre-weaning PD on neuronal activities of PrL, NAc (the number of Fos-IR cells) and PVN (the percentage of c-fos/OT double-labeled neurons) while exposure to a juvenile vole have also been investigated. We believe the manuscript have been improved significantly compared with last version.

Reviewer #1:This findings from this study show that paternal deprivation during the pre-weaning period (PD14-21) of mandarin voles leads to increased anxiety-like behavior and diminished social preferences and implicate disruption of the mPFC OT system in these effects. Overall the question addressed is interesting and the results are straightforward although I do have several comments/concerns:1) The authors suggest that the detrimental effects of PD are due to disrupted "emotional attachment" but they provide little if any justification for this. Showing increases in anxiety behavior and attenuated social preference is not evidence that disrupted attachment is the casual factor.

One of our previous studies found that mandarin vole pups display stronger emotional attachment towards fathers by social preference test from 14 to 21 days of age (He et al., 2017). In the present study, pre-weaning paternal deprivation increased anxiety behavior and attenuated social preference. Thus, we suggest that disruption of emotional attachment between fathers and offspring causes these effects.

2) The use of OTR receptor antibodies has been called into question. Therefore, more details should be provided about the specificity of this antibody and any control experiments that were performed. Ideally, the authors would corroborate the current data with another method.

According the reviewer’s suggestion, positive (mouse spleen tissue lysate) control experiment was performed. The result further supports the specificity of this antibody.

**Author response image 1. respfig1:** Lane 2 and Lane 3: Anti-Oxytocin Receptor antibody [EPR12789] (ab181077, Abcam) at 1/2000 dilution. Lane 1: Protein marker. Lane 2: Voles mPFC tissue lysate. Lane 3: Mouse spleen tissue lysate.

3) In various places throughout the manuscript the authors discuss the effects of OT on the V1a receptor and even suggest in the Discussion that OT may regulate anxiety-like behavior and social preference in males via OTR and V1aR in the cortex. Thus, it seems surprising that in the pharmacology experiment (experiment 3) they didn't include an additional group that was co-administered OT with an AVP receptor antagonist.

As reviewer suggested, an additional group that was co-administered OT with an AVP receptor antagonist (10ng OT/10ng V1aRA) group has been included.

4) The description of the statistical results for experiment one is inaccurate. Specifically, the% Investigation data for males and females are described the same way except that for males there was an interaction and for females there was only a main effect of stimulus.

Based on reviewer’s suggestions, we have changed the description of the statistical results.

“No significant treatment x stimulus was found on the percentage of investigation time in females (F(1,24) =1.553, p = 0.225), but a main effect of absence/presence of a stimulus mandarin vole was found (F(1,24) = 11.785, p < 0.01)”.

5) For experiment 3, the possibility of OT diffusion beyond the PL mPFC wasn't addressed. Also, it was mentioned that data were excluded if cannula tips were located in other brain regions. I wonder if the number of subjects that had missed cannula placements is sufficient to form a "miss group" as a control for site specificity.

It is a good suggestion. But “miss group” only have one vole in male group (Male: 10ng OT /100 ng OTA: n = 5) and two voles in female group (Female: 1ng OT /10 ng OTA: n = 5, 1ng OT /100 ng OTA: n = 5). The sample size is too small to for a “miss group”. Furthermore, OT only in the PL mPFC decreased anxiety-like behavior, but not in the Cg and IL (Sabihi et al., 2017).

Sabihi S, Dong SM, Maurer SD, Post C, Leuner B. 2017. Oxytocin in the medial prefrontal cortex attenuates anxiety: anatomical and receptor specificity and mechanism of action. Neuropharmacology 125:1–12.

Reviewer #2:[…] The weakness of the report is the lack of novelty. Oxytocin is arguably the most intensely investigated peptide on the planet for its role in social bonding and anxiety. Indeed the authors make such a compelling argument that the anxiety induced by the loss of a parent must be mediated by oxytocin that any finding to the contrary would have been truly surprising, and interesting. But as it stands, the current report only modestly adds to the existing and extensive literature.

Early life stress such as disruption of parents-offspring bonds is closely associated with pathology of adult neuropsychiatric disorders. Most research in rodents has focused on the effects of early life stress using maternal separation from postnatal day (PND) 0 to 14. In contrast, the effects of paternal deprivation (PD) on adult behavior and brain are still poorly understood. The neural mechanisms underlying effects of disruption of the pre-weaning father-offspring attachment on offspring behavior remain elusive and circuit mechanisms are poorly understood. Our result revealed that pre-weaning PD increases anxiety and diminishes social preferences. The prelimbic cortex OT system may be involved in this process. For first time, we found that PD-induced anxiety-like behavior and social preference impairment could be rescued by optogenetics activation of the PVN-PrL OTergic fibers. Altogether, these results provide a new pathway via which PD possibly affects emotion and social behavior, and suggest a critical target for treatment of mental disorders associated with PD.

What would be an advance would be some mechanistic insight into how oxytocin innervation is impacted by early life depravation of a parent. Do the oxytocin neurons die? Or is expression of the gene repressed? Are there enduring epigenetic modifications? Etc. However before that line of inquiry were to be pursued it would also be important to provide far more convincing evidence that the innervation has in fact changed. The images provided here are not reassuring from a quantitative standpoint.

We found that PD decreased the number of OT-positive neurons in the PVN. Some studies found that the prefrontal cortex is much more affected than other brain areas after maternal separation (Gapp et al., 2016). We speculate that oxytocinergic neurons within the PVN project to the PrL is impacted by PD. Furthermore, some genes and epigenetic modifications need further study. Because there are a few OT-immunoreactive ﬁbers in the PrL and we do not provide more convincing evidence, we have deleted the result of OT fibers.

Gapp, K., Corcoba, A., Van Steenwyk, G., Mansuy, I. M., and Duarte, J. M. (2016). Brain metabolic alterations in mice subjected to postnatal traumatic stress and in their offspring. Journal of Cerebral Blood Flow and Metabolism, 37, 2423–2432.

Reviewer #3:[…] In sum, the novelty of the present manuscript is limited to a specific paradigm of PD of Mandarin voles at PND 14-21. Moreover, the manuscript does not provide sufficient data regarding the mechanism of the behavioral and neuroanatomical deficits. The behavioral rescue with OT injection (Figure 7) is somewhat novel but very limited and preliminary in terms of identifying the exact neuronal circuit which is affected by OT administration. I also have some major technical concerns which are listed below.

To further investigate underlying mechanisms underlying effects of pre-weaning paternal deprivation, in the new version of the manuscript, we have added new experiment to investigate the effect of pre-weaning PD on neuronal activities of PrL, NAc (the number of Fos-IR cells) and PVN (the percentage of c-fos/OT double-labeled neurons) while exposure to a juvenile vole. In addition, we investigated involvement of PVN OT neurons projection to mPFC in effects of paternal deprivation using optogenetic methods. For first time, we found that activation of oxytocinergic terminals in the mPFC by optogenetic method rescued social preference and decreased anxiety-like behavior induced by paternal deprivation.

Specific major concerns:Figure 1: Additional anxiety-related behavioral assays (i.e. elevated plus maze, Dark-light transfer assay) would assist in further characterization of anxiety in PD voles, and could contribute to establishing the differential response to PD in females vs. males.

We added the Light-Dark box test to assess anxiety-like behavior for adult voles. No signiﬁcant sex x treatment interaction was found on the percentage of time in the central area (F(1,24) = 0.019, p = 0.893) and the percentage of time in light area (F(1,24) = 0.039, p = 0.884). Treatments also produced signiﬁcant eﬀects on the percentage of time in the central area (F(1,24) = 25.628, p < 0.01) and the percentage of time in light area (F(1,24) = 19.812, p < 0.01), whereas sex did not produce a signiﬁcant main eﬀect in the percentage of time in the central area (F(1,24) = 0.029, p = 0.867) and the percentage of time in light area (F(1,24) = 0.227, p = 0.638). Thus, the data from the diﬀerent sexes were analyzed separately.

Figure 3: The images showing the effect of PD on PLC OT-IR fibers are of very poor quality and not convincing. The reader will find it hard to believe that hundreds of OT-IR fibers were reliably quantified.

We have deleted this section, and we have added experimental result regarding to effect of pre-weaning PD on PrL, NAc (the number of Fos-IR cells) and PVN (the percentage of c-fos/OT double-labeled neurons) neuronal activities after exposure to a juvenile vole compared to exposure to a toy.

Figure 4: This figure is all about a negative result (no difference in nucleus accumbens OT fibers) and it is not necessary to show as a main figure. It will be more appropriate as either a supplementary figure or as part of Figure 3.

We found that PD decreased the number of OT-positive neurons in the PVN. Some studies found that the prefrontal cortex is much more affected than other brain areas after maternal separation (Gapp et al., 2016). We speculate that oxytocinergic neurons within the PVN project to the PrL is impacted by PD. Because there are a few OT-immunoreactive ﬁbers in the PrL and we do not provide more convincing evidence, we have deleted the result of OT fibers in the PrL and NAc.

Gapp, K., Corcoba, A., Van Steenwyk, G., Mansuy, I. M., and Duarte, J. M. (2016). Brain metabolic alterations in mice subjected to postnatal traumatic stress and in their offspring. *Journal of Cerebral Blood Flow and Metabolism*, 37, 2423–2432.

Figure 5: The quality of the Western blots showing the levels of OTR and V1aR is unacceptable. Firstly, the gels are cropped so the specificity of the antibodies (which are known to be problematic) cannot be appreciated. Secondly, some of the protein bands are so weak and close to the background level, at some panels and the loading controls of anti-Tubulin are not equal and therefore the experimental evidences regarding the effects of PD on OTR (Figure 5A and B) and V1aR (Figure 5C) are not sufficiently supported.

As reviewer suggested, more additional Western blots experiments have been done in order to obtain more accurate result.

Figure 6: This figure demonstrates the effects of PD on OT serum levels in females. Measuring OT serum levels using ELISA kits is criticized in the field [see: Leng and Sabatier J Neuroendocrinol. 2016 Oct;28(10). doi: 10.1111/jne.12413]. In particular, the method of extraction is critical and may cause artifacts. The authors state that this was performed using "vole-specific enzyme-linked immunosorbent assay (Shanghai Xitang Biotechnology, Shanghai, China), according to the manufacturers' instructions." however, they do not provide details or a reference regarding the validity of this assay.

Thank you for your suggestions. Measuring serum OT levels may be lacked in reliability when using EIA-kits. These high values may be due to they lack in reliability when determining OT levels in unextracted samples of bodily fluids, keeping in mind that there will be more high molecules tagged than OT alone (Leng and Sabatier, 2016). When these interfering factors were removed by extraction of plasma samples, immunoassays gave measurements consistent with bioassays. However, many recent papers use an enzyme-linked immunoassay to measure plasma levels without extracting the samples. Like the first radioimmunassays of unextracted plasma, this generates impossibly high and wholly erroneous measurements (Leng and Sabatier, 2016). Thus, we have deleted the result of OT and CORT serum levels.

Leng G, Sabatier N. 2016. Measuring oxytocin and vasopressin: bioassays, immunoassays and random numbers. Journal of Neuroendocrinology28. Doi: 10.1111/jne.12413.

Figure 7: Effects of PLC OT administration on anxiety and social preference behaviors of PD voles. The presentation of data in this figure is very confusing. In particular, it is not clear whether the authors compared the behavioral effects of OT on PD vs. naïve (non-PD) voles. This should be clarified.

Because there were six groups in the experiment divided into, and significant differences were found among many groups. Thus, we used “Bars without same letters are signiﬁcantly different” in the Figure 7. We have clarified that in the figure captions “Effects of PLC OT administration on anxiety-like behavior and social preference of PD mandarin voles”.

[Editors' note: the author responses to the re-review follow.]

Summary:The reviewers agreed that the additional experiments carried out have strengthened this paper, overcoming several of the issues that were raised previously, as well as providing a more direct link between the effects of parental deprivation and activity in the projection from oxytocinergic PVN neurons to the mPFC. They differed, however, in their views about the significance of these findings.The reviewers' comments are included below. In addition, the following points have been made by the Reviewing Editor and need to be addressed in any revision:1) The optogenetic experiments require further information and important controls are missing. These include making injections that omit ChR2 from the viral construct to demonstrate the specificity of the effect; quantifying the co-localization of ChR2-mCherry and oxytocin expression in the PVN;

As reviewer suggested, more details about the optogenetic experiments were added to the Materials and methods and the Results. The viral construct that omits the ChR2 has been injected and specificity of the effects has been demonstrated.

ChR2-mCherry expression neurons coexpressed OT have been quantified and it is found that the majority of neurons were ChR2 and OT coexpressed. In male, percentage of ChR2 and OT coexpressed neuron in ChR2 is 60.6% (132/218, from 2 voles) and that in female is 70.9% (190/268, from 2 voles).

We have added data about colocalization of OT/mCherry of PD voles (Figure 7—figure supplement 3) and naive voles (Figure 8—figure supplement 1).

Providing more details about the overall duration over which optogenetic stimulation was delivered.

Detailed information has been added in the Materials and methods. During optogenetic experiments, the durations of optogenetic stimulation in both the social preference test and social memory were 10 min based on one previous study (Hung et al., 2017). Thus, subjects received about 25 minutes blue laser pulses during whole test phase.

How long do the behavioral changes last following optogenetic activation?

Blue light photostimulation produced obvious changes in behavior and this change disappeared 8 hours after photostimulation in PD voles (Figure 7—figure supplement 4). The naïve voles still exhibited changes in anxiety-like behavior and decreased social preference (except male) 8 hours after yellow light illumination, the effect of yellow light stimulation disappeared within 8-24hours (Figure 8—figure supplement 2).

This is particularly important if you wish to argue that abnormal social behavior can be rescued in this fashion, and relates to the concern raised by reviewer 1 about the permanence (or otherwise) of the effects of parental deprivation.

We added the further discussion according to the reviewer’s suggestions.

“Changes in OT pathways induced by severe early life stress can be transmitted to the next generation (Toepfer et al., 2017). […] Thus, parental deprivation impacts the offspring oxytocin system on permanent basis.”

Can the behavioral changes be rescued by providing PD voles with increased social interactions in later life?

Because amicable social interaction may increase release of oxytocin and change the OT system, and possibly rescue the behavioral changes induced by PD with involvement of OT system. It is good suggestion. We can test that in the future study.

The optogenetic experiments would also be strengthened if the behavioral consequences of expressing an inhibitory opsin in the PVN-mPFC axons were measured.

As reviewer suggested, the naive voles were injected with virus with promotor gene of inhibitory opsin (rAAV-Ef1α-DIO-eNpHR3.0-mCherry+ rAAV-Oxytocin-Cre) or control virus (rAAV-Ef1α-DIO-mCherry+ rAAV-Oxytocin-Cre) into the bilaterally PVN. We also found that optogenetic inhibition of the PVN (OTergic)-PrL pathway impaired the social preference and increased levels of anxiety-like behavior in the naïve voles displaying the effects of preweaning PD (Figure 8).

2) The quality of the writing is poor and in places does not make sense. The meaningless expression 'socially mandarin vole' is still in the Abstract. The reference list is also incomplete.

As reviewer suggested, the manuscript has been edited by a native English editor. We have deleted “socially mandarin voles” in the Abstract and checked the reference one by one.

3) There are many plots in which histograms with error bars are used; individual data points should be shown rather than error bars.

Based on the reviewer’s suggestion, we have added individual data points in histograms.

Reviewer #1:The authors have done some additional work and corrected some weaknesses but my initial concern was that the finding was predictable and so novelty is low. Understanding how parental deprivation impacts the oxytocin system on a permanent basis would be an advance. That advance is not made here.

We added further discussion according to the reviewer’s suggestions.

“Changes in OT pathways induced by severe early life stress can be transmitted to the next generation (Toepfer et al., 2017). […] Thus, parental deprivation impacts the offspring oxytocin system on permanent basis.”